# THE JOURNEY, NOT THE DESTINATION: HOW DATA GUIDES DIFFUSION MODELS

## ABSTRACT

Diffusion models trained on large datasets can synthesize photo-realistic images of remarkable quality and diversity. However, *attributing* these images back to the training data—that is, identifying specific training examples which *caused* an image to be generated—remains a challenge. In this paper, we propose a framework that: (i) provides a formal notion of data attribution in the context of diffusion models, and (ii) allows us to *counterfactually* validate such attributions. Then, we provide a method for computing these attributions efficiently. Finally, we apply our method to find (and evaluate) such attributions for denoising diffusion probabilistic models trained on CIFAR-10 and latent diffusion models trained on MS COCO.

## 1 INTRODUCTION

Diffusion models can generate novel images that are simultaneously photorealistic and highly controllable via textual prompting (Ramesh et al., 2022; Rombach et al., 2022). A key driver of diffusion models' performance is training them on massive amounts of data (Schuhmann et al., 2022). Yet, this dependence on data has given rise to concerns about how diffusion models use it.

For example, Carlini et al. (2021); Somepalli et al. (2022) show that diffusion models often memorize training images and "regurgitate" them during generation. However, beyond such cases of direct memorization, we currently lack a method for *attributing* generated images back to the most influential training examples—that is, identifying examples that *caused* a given image to be generated. Indeed, such a primitive—a *data attribution method*—would have a number of applications. For example, previous work has shown that attributing model outputs back to data can be important for debugging model behavior (Shah et al., 2022), detecting poisoned or mislabelled data (Lin et al., 2022), and curating higher quality training datasets (Khanna et al., 2019). Within the context of diffusion models, data attribution can also help detect cases of data leakage (i.e., privacy violations), and more broadly, can be a valuable tool in the context of tracing content provenance relevant to questions of copyright (Andersen et al., 2023; Images, 2023). Finally, synthetic images generated by diffusion models are now increasingly used across the entire machine learning pipeline, including training (Azizi et al., 2023) and model evaluation (Kattakinda et al., 2022; Wiles et al., 2022; Vendrow et al., 2023). Thus, it is critical to identify (and mitigate) failure modes of these models that stem from training data, such as bias propagation (Luccioni et al., 2023; Perera & Patel, 2023) and memorization. Motivated by all the above needs, we thus ask:

*How can we reliably attribute images synthesized by diffusion models back to their training data?*

Although data attribution has been extensively studied in the context of *supervised* learning (Koh & Liang, 2017; Ghorbani et al., 2019; Jia et al., 2019; Ilyas et al., 2022; Hammoudeh & Lowd, 2022; Park et al., 2023), the generative setting poses new challenges. First, it is unclear *what* particular behavior of these models we hope to attribute. For example, given a generated image, certain training images might be responsible for the look of the background, while others might be responsible for the choice of an object appearing in the foreground. Second, it is not immediately obvious how to *verify* the attributions, i.e., how to compare the outputs of the original model with those of a new model trained on a new dataset after removing the attributed examples.

**Our contributions.** In this work, we present a data attribution framework for diffusion models. This framework reflects, and is motivated by, the fact that diffusion models iteratively denoise an

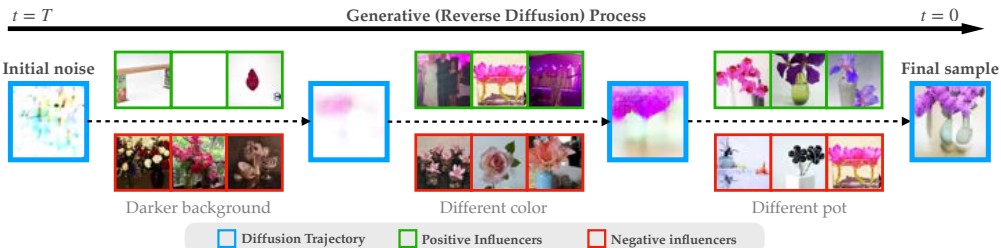

Figure 1: **Overview of our attribution method.** For a given synthesized image, we apply our attribution method at individual steps along the diffusion trajectory. At each step $t$, our method pinpoints the training examples with the highest influence (positive in green, negative in red) on the generative process at this step. In particular, positive influencers guide the trajectory towards the final sample, while negative influencers guide the trajectory away from it. We observe that negative influencers increasingly resemble the final sample (the grey text highlights notable differences). For more examples, see Appendix C.5.

initial random seed to reach a final generated image. In particular, rather than attributing *only* the final generated image, i.e., the "destination," we attribute each individual step along the (denoising) "journey" taken by diffusion model (see Figure 1); this approach enables us to surface attributions targeted towards specific features of the final generated image. We then introduce two complementary metrics for evaluating the resulting attributions based on measuring distribution-wide rather than sample-specific changes. Finally, we provide an efficient method for computing attributions within this framework, building on data attribution approaches developed for the supervised setting (Ilyas et al., 2022; Park et al., 2023). We apply our method to denoising diffusion probabilistic models (DDPM) (Ho et al., 2020) trained on CIFAR-10 (Krizhevsky, 2009), and latent diffusion models (LDM) (Rombach et al., 2022) trained on MS COCO (Lin et al., 2014). In both of these settings, we obtain attributions that are validated by our metrics and also visually interpretable.

## 1.1 RELATED WORK

In a concurrent work, Wang et al. (2023) propose a method for *efficiently evaluating* data attribution methods for generative models by creating custom datasets with known ground-truth attributions. Additionally, recent works have studied *memorization*, which can be thought of as a special case of data attribution where only few, nearly identical images in the training set are responsible for the generation of a corresponding image. In particular, Somepalli et al. (2022); Carlini et al. (2023) use image similarity metrics ($\ell_2$ distance in pixel space and CLIP embeddings) to pinpoint cases of memorization in diffusion models. In Appendix B, we discuss related work in more depth.

## 2 PRELIMINARIES

We first provide background on data attribution. Then, we give a brief overview of diffusion models, highlighting the components that we will need to formalize attribution for these models.

**Data attribution.** Broadly, the goal of training data attribution (Koh & Liang, 2017; Ilyas et al., 2022; Hammoudeh & Lowd, 2022; Park et al., 2023) is to trace model outputs back to the training data. Intuitively, we want to estimate how the presence of each example in the training set impacts a given model output of interest (e.g., the loss of a classifier) on a specific input. To formalize this, consider a learning algorithm $\mathcal{A}$ (e.g., a training recipe for a model), together with an input space $\mathcal{Z}$ and a training dataset $S = (z_1, \ldots, z_n) \in \mathcal{Z}^n$ of $n$ datapoints from that input space. Given a datapoint $z \in \mathcal{Z}$, we represent the model output via a *model output function* $f(z, \theta(S)) : \mathcal{Z} \times \mathbb{R}^d \to \mathbb{R}$, where $\theta(S) \in \mathbb{R}^d$ denotes the model parameters resulting from running algorithm $\mathcal{A}$ on the dataset $\mathcal{S}$. For example, $f(z, \theta(S))$ is the loss on a test sample $z$ of a classifier trained on $S$. ( Our notation here reflects the fact that the parameters are a function of the training dataset $S$.) We now define a *data attribution method* as a function $\tau : \mathcal{Z} \times \mathcal{Z}^n \to \mathbb{R}^n$ that assigns a score $\tau(z, S)_i \in \mathbb{R}$ to each training

example $z_i \in S$.[1] Intuitively, we want $\tau(z, S)_i$ to capture the change in the model output function $f(z, \theta(S))$ induced by adding $z_i$ to the training set.

More generally, these scores should help us make *counterfactual* predictions about the model behavior resulting from training on an arbitrary subset $S' \subseteq S$ of the training datapoints. We can formalize this goal using the *datamodeling* task Ilyas et al. (2022): given an arbitrary subset $S' \subseteq S$ of the training set, the task is to predict the resulting model output $f(z, \theta(S'))$. A simple method to use the attribution scores for this task, then, is to consider a *linear* predictor: $f(z, \theta(S')) \approx \sum_{i:z_i \in S'} \tau(z, S)_i$.

**Estimating attribution scores (efficiently).** Given the model output function $f$ evaluated at input $z$, a natural way to assign an attribution score $\tau(z)_i$ for a training datapoint $z_i$ is to consider the *marginal* effect of including that particular example on the model output, i.e., have $\tau(z)_i = f(z, \theta(S)) - f(z, \theta(S \setminus \{z_i\}))$. We can further approximate this difference by decomposing it as:

$$\tau(z)_i = \underbrace{(\theta - \theta_{-i})}_{\text{(i) change in model parameters}} \cdot \overbrace{\nabla_\theta f(z, \theta)}^{\text{(ii) change in model output}}, \tag{1}$$

where $\theta_{-i}$ denotes $\theta(S \setminus \{i\})$ (Wojnowicz et al., 2016; Koh & Liang, 2017). We can compute the second component efficiently, as this only requires taking the gradient of the model output function with respect to the parameters; in contrast, computing the first component is not always straightforward. In simpler settings, such as linear regression, we can compute the first component explicitly, as there exists a closed-form solution for computing the parameters $\theta(S')$ as a function of the training set $S'$. However, in modern, non-convex settings, estimating this component efficiently (i.e., without retraining the model) is challenging. Indeed, prior works such as influence functions (Koh & Liang, 2017) and TracIn (Pruthi et al., 2020) estimate the change in model parameters using different heuristics, but these approaches can be inaccurate in such settings.

To address these challenges, TRAK (Park et al., 2023) observed that for deep neural networks, approximating the original model with a model that is *linear* in its parameters, and averaging the estimates over multiple $\theta$'s (to overcome stochasticity in training) yields highly accurate attribution scores. The linearization is motivated by the observation that at small learning rates, the trajectory of gradient descent on the original neural network is well approximated by that of a corresponding linear model (Long, 2021; Wei et al., 2022; Malladi et al., 2022). In this paper, we will leverage the TRAK framework towards attributing diffusion models.

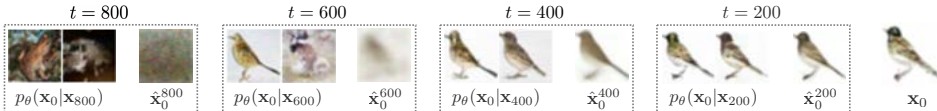

Figure 2: Samples from $p_\theta(\cdot|\mathbf{x}_t)$, i.e., the distribution of final images $\mathbf{x}_0$ conditioned on initializing from the latent $\mathbf{x}_t$ at step $t$, and the corresponding approximation $\hat{\mathbf{x}}_0^t$ (a proxy for the expectation of this distribution, i.e., $\mathbb{E}[\mathbf{x}_0|\mathbf{x}_t]$) for different values of $t$, together with the final generated image $\mathbf{x}_0$.

**Diffusion models.** At a high level, diffusion models (and generative models, more broadly) learn a distribution $p_\theta(\cdot)$ meant to approximate a target distribution $q_{data}(\cdot)$ of interest (e.g., natural images). Given a sample $\mathbf{x}_0 \sim q_{data}(\cdot)$, diffusion models leverage a stochastic *diffusion* process that gradually corrupts $\mathbf{x}_0$ by adding noise at each step, obtaining a sequence of samples $\{\mathbf{x}_t\}_{t \in [T]}$ where $\mathbf{x}_t \sim \mathcal{N}(\alpha_t \cdot \mathbf{x}_{t-1}, (1 - \alpha_t) \cdot I)$[2] (Sohl-Dickstein et al., 2015; Song & Ermon, 2019; Ho et al., 2020). Then, during training, these models learn a neural network $\varepsilon_\theta$ that runs this process in reverse. Given an initial seed $\mathbf{x}_T \sim \mathcal{N}(0, 1)$, we apply the network iteratively at each step $t$ (from $t = T$ to $t = 0$) to sample the *diffusion trajectory* $\{\mathbf{x}_t\}_{t \in [T]}$, ultimately leading to a final sample $\mathbf{x}_0 \sim p_\theta(\cdot) \approx q_{data}(\cdot)$.

In this work, it will also be useful to consider the process of sampling a final image $\mathbf{x}_0$ when "resuming" the diffusion process after running it up to some step $t$—this is equivalent to continuing the denoising

---

[1]Following the literature, we say that an example $z_i$ has a *positive (respectively, negative) influence* if $\tau(z, S)_i > 0$ (respectively, $\tau(z, S)_i < 0$).

[2]Here, $\{\alpha_t\}_t$ are parameters of the diffusion process.

process at step $t$ from the corresponding intermediate latent $\mathbf{x}_t$. We denote the distribution arising from sampling an image $\mathbf{x}_0$ when conditioning on the latent $\mathbf{x}_t$ by $p_\theta(\cdot|\mathbf{x}_t)$.

It turns out that we can approximate the multi-step denoising process of generating samples from $p_\theta(\cdot|\mathbf{x}_t)$ in a single step with the formula $\hat{\mathbf{x}}_0^t := c_1(\alpha_t) \cdot (\mathbf{x}_t - c_2(\alpha_t \cdot \boldsymbol{\varepsilon}_\theta(\mathbf{x}_t, t)))$, for some constants $c_1(\cdot), c_2(\cdot)$ depending on the diffusion parameters $\{\alpha_t\}_t$ (Ho et al., 2020). In fact, $\hat{\mathbf{x}}_0^t$ is a proxy for the conditional expectation $\mathbb{E}[\mathbf{x}_0|\mathbf{x}_t]$, i.e., the expectation of $p_\theta(\cdot|\mathbf{x}_t)$, and under certain conditions $\hat{\mathbf{x}}_0^t$ is precisely equivalent to this expectation.[3] See Figure 2 for an illustration of $p_\theta(\cdot|\mathbf{x}_t)$ and $\hat{\mathbf{x}}_0^t$ for different values of $t$.

Recently, Rombach et al. (2022) proposed a new class of diffusion models called latent diffusion models (LDMs), which perform the above stochastic process in the latent space of a pretrained encoder network. Moreover, as Song et al. (2021); Ho & Salimans (2022) have shown, one can also *condition* diffusion models on some additional information, e.g. a text prompt. This way, one can explicitly control the semantics of the generated images via the choice of such a text prompt.

## 3    A DATA ATTRIBUTION FRAMEWORK FOR DIFFUSION MODELS

In this section, we introduce our framework for attributing samples generated by diffusion models. To this end, we specify both *what* to attribute as well as how to *verify* the attributions. In particular, in Section 3.1 we define data attribution for diffusion models as the task of understanding how training data influence the distribution over the final images at *each* step of the diffusion process. Then, in Section 3.2, we describe how to evaluate and verify such attributions.

### 3.1    ATTRIBUTING THE DIFFUSION PROCESS STEP BY STEP

Diffusion models generate images via a *multi-step* process. We thus choose to decompose the task of attributing a final synthesized image into a series of stages too, each stage providing attributions for a single step of the overall diffusion process. Such attribution at the level of individual steps allows for:

1. **Fine-grained analysis.** Identifying influential training examples at each individual step enables us to gain a fine-grained understanding of how data *guides* the diffusion process This allows us to capture, for example, that in some cases the same image might be positively influential at early steps but negatively influential at later steps (see Appendix C.2).

2. **Feature-level attribution.** As demonstrated below, attributions at an individual step level allow us to isolate influences on individual features within a final generated image.

3. **Computational feasibility.** Computing gradients through a single step requires only a single backwards pass. So, it becomes feasible to apply existing efficient data attribution methods (Park et al., 2023; Pruthi et al., 2020) that involve computing gradients.

It remains now to define what exactly to attribute to the training data at each step. To this end, we first motivate studying the conditional distribution $p_\theta(\cdot|\mathbf{x}_t)$ (see Section 2) as a way to quantify the impact of each step $t$ of the diffusion process to the final sample $\mathbf{x}_0$. Next, we highlight how analyzing the evolution of this distribution over steps $t$ can connect individual steps to specific features of interest. Finally, building on these observations, we formalize our framework as attributing properties of this distribution $p_\theta(\cdot|\mathbf{x}_t)$ at each step $t$ to the training data.

**Studying the distribution** $p_\theta(\cdot|\mathbf{x}_t)$**.** At a given step $t$ of the generative process, the relevant information about the process up to that point is contained in the latent $\mathbf{x}_t$. While $\mathbf{x}_t$ itself might not correspond to a natural image, we can use it to directly sample from $p_\theta(\cdot|\mathbf{x}_t)$, i.e., the distribution of possible final images $\mathbf{x}_0$ when resuming the diffusion process at step $t$ with latent $\mathbf{x}_t$. When $t = T$, this distribution is precisely the diffusion model's learned distribution $p_\theta(\cdot)$, and at $t = 0$ it is simply the final sampled image $\mathbf{x}_0$. So, intuitively, the progression of this conditional distribution over steps $t$ informs us how the model gradually "narrows down" the possible distribution of samples to generate the final sample $\mathbf{x}_0$ (see Figure 2 for an illustration). A natural way to understand (and attribute) the impact of applying the diffusion process at each step $t$ on the final image $\mathbf{x}_0$ is thus to understand how this conditional distribution $p_\theta(\cdot|\mathbf{x}_t)$ evolves over steps.

---

[3]This equivalence is referred to as the *consistency* property (Song et al., 2023; Daras et al., 2023).

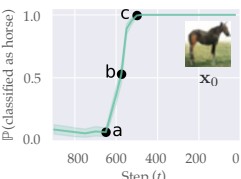 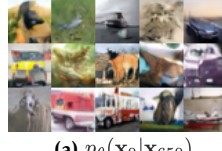 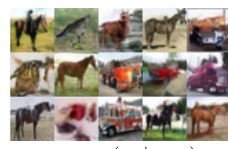 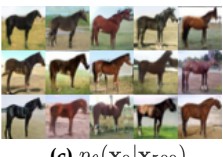

**(a)** $p_\theta(\mathbf{x}_0|\mathbf{x}_{650})$     **(b)** $p_\theta(\mathbf{x}_0|\mathbf{x}_{575})$     **(c)** $p_\theta(\mathbf{x}_0|\mathbf{x}_{500})$

Figure 3: **Specific features appearing at specific steps. (Left)** For a given image of a horse ($\mathbf{x}_0$) generated by a CIFAR-10 DDPM model, we plot the likelihood that samples from the distribution $p_\theta(\cdot|\mathbf{x}_t)$ (see Section 2) are classified as a horse according to a CIFAR-10 classifier. This likelihood increases rapidly around steps 650 to 500, suggesting that these steps are most responsible for the formation of this feature. **(Top)** For three steps $t$ in this range, we visualize samples from $p_\theta(\cdot|\mathbf{x}_t)$. **(Bottom)** At each of these steps, we also visualize the training examples with the highest influence (positive in green, negative in red) identified by our method. Note that once the "horse" feature begins to appear (around $t = 575$), positive influencers begin to reflect it. However, after this feature is "decided" (around $t = 500$), negative influencers *also* begin to reflect it.

**Connecting features to specific steps.** Given a final generated image, there might be many possible *features* of interest within this image. For example, for $\mathbf{x}_0$ in Figure 2, we might ask: *Why is there a grey bird? Why is the background white?* How can we quantify the impact of a particular step $t$ on a given feature in the final image? To answer this question, we simply sample from the conditional distribution $p_\theta(\cdot|\mathbf{x}_t)$ and measure the fraction of samples that contain the feature of interest. Now, if we treat this (empirical) likelihood as a function of $t$, the steps at which there is the largest increase in likelihood are most responsible for the presence of this feature in the final image.

In fact, it turns out that such rapid increase in likelihood often happens within only a small interval; we observe this phenomenon for both small-scale unconditional models (DDPM trained on CIFAR-10, Figure 3) and large-scale text-conditional models (Stable Diffusion v2 trained on LAION-5B, Appendix C.4). As a result, we are able to tie the presence of a given feature in the final image back to a small interval of steps $t$ in the sampling process.

**Implementing our approach.** To implement our step-by-step attribution approach, we need a model output function (see Section 2) that is specific to a step $t$. As we motivated above, this function should be applied to samples from the conditional distribution $p_\theta(\cdot|\mathbf{x}_t)$. To that end, we introduce a step-specific model output function $f_t(p_{\theta(S)}(\cdot|\mathbf{x}_t), \theta(S))$. The function $f_t$ is intended to measure properties of the distribution $p_{\theta(S)}(\cdot|\mathbf{x}_t)$. For example, in Section 4 we define a concrete instantiation of $f_t$ that approximates the likelihood of the model to generate individual samples from $p_{\theta(S)}(\cdot|\mathbf{x}_t)$. Adapting the general definition of data attribution from Section 2, we can now define *data attribution for diffusion models* at a step $t$ as a function $\tau_t$ that assigns a score $\tau_t(\mathbf{x}_t, S)_i$ to each training example $z_i \in S$. This score indicates the change in $f_t(p_{\theta(S)}(\cdot|\mathbf{x}_t), \theta(S))$ induced by adding $z_i$ to $S$.

## 3.2 VALIDATING DATA ATTRIBUTION FOR DIFFUSION MODELS

Visually inspecting the attributed training examples is a common heuristic for evaluating data attribution, but visual similarity is not always reliable (Ilyas et al., 2022; Park et al., 2023). In particular, applications of data attribution (such as data curation) often require that the attributions are causally predictive. To that end, we evaluate attribution scores according to how accurately they reflect the corresponding training examples' *counterfactual* impact on the conditional distribution $p_\theta(\cdot|\mathbf{x}_t)$ using two different metrics. The first metric, the linear datamodeling score, considers models trained on random subsets of the full training set, whereas the second metric considers models trained on specific counterfactual training sets targeted for each generated image. The first metric is cheaper to evaluate as we can re-use the same set of models to evaluate different target images.

**Linear datamodeling score.** The linear datamodeling score (LDS) is a measure of the effectiveness of a data attribution method that was introduced in Ilyas et al. (2022); Park et al. (2023) (see Section 2). This metric quantifies how well the attribution scores can predict the exact *magnitude* of change in model output induced by (random) variations in the training set. In our setting, we use the attribution

scores $\tau$ to predict the diffusion-specific model output function $f_t(p_{\theta(S)}(\cdot|\mathbf{x}_t), \theta(S))$ as

$$g_\tau(p_\theta(\cdot|\mathbf{x}_t), S'; S) := \sum_{i\,:\,z_i \in S'} \tau(\mathbf{x}_t, S)_i = \tau(\mathbf{x}_t, S) \cdot \mathbf{1}_{S'}, \qquad (2)$$

where $\mathbf{1}_{S'}$ is the *indicator vector* of the subset $S'$ of $S$ (i.e., $(\mathbf{1}_{S'})_i = \mathbf{1}\{z_i \in S'\}$). Then, we can measure the degree to which the predictions $g_\tau(p_\theta(\cdot|\mathbf{x}_t), S'; S)$ are correlated with the true outputs $f_t(p_\theta(\cdot|\mathbf{x}_t), \theta(S'))$ using the LDS:

$$LDS(\tau, \mathbf{x}_t) := \boldsymbol{\rho}(\{f_t(p_\theta(\cdot|\mathbf{x}_t), \theta(S_j)) : j \in [m]\}, \{g_\tau(p_\theta(\cdot|\mathbf{x}_t), S_j; S) : j \in [m]\}),$$

where $\{S_1, \ldots, S_m : S_i \subset S\}$ are randomly sampled subsets of the training set $S$ and $\boldsymbol{\rho}$ denotes Spearman's rank correlation (Spearman, 1904). To decrease the cost of computing LDS, we use $\hat{\mathbf{x}}_0^t$ in lieu of samples from $p_\theta(\cdot|\mathbf{x}_t)$. As we noted in Section 2, this is a proxy for the conditional expectation $\mathbb{E}[\mathbf{x}_0|\mathbf{x}_t]$. In other words, we consider $f_t$ and $g_\tau$ as functions of $\hat{\mathbf{x}}_0^t$ rather than $p_\theta(\cdot|\mathbf{x}_t)$.

**Retraining without the most influential images.**    In practice, we may want to use the attributions to intentionally steer the diffusion model's output; for example, we may want to remove all training examples that cause the resulting model to generate a particular style of images. To evaluate the usefulness of attribution method in these contexts, we remove from the training set the most influential (i.e., highest scoring) images for a given target $\mathbf{x}_t$, retrain a new model $\theta'$, then measure the change in the conditional distribution $p_{\theta(S)}(\cdot|\mathbf{x}_t)$ when we replace $\theta$ with $\theta'$ only in the neighborhood of time step $t$ in the reverse diffusion process. If the attributions are accurate, we expect the conditional distribution to change significantly, which we measure using the FID distance (Heusel et al., 2017).

As we consider attributions specific to each time step, in principle we should use the denoising model *only* for the corresponding step $t$. However, the effect of a single time step on the final distribution may be small and hard to quantify. Hence, we assume that attributions change gradually over time and replace the denoising model for a *small interval* of time steps (i.e., between steps $t$ and $t - \Delta$).

## 4 METHODS

In this section, we describe how we estimate attributions for diffusion models by adapting the data attribution method TRAK Park et al. (2023) (see Section 2 for background on TRAK).

**Estimating the change in model parameters.**    For diffusion models, the training process is much more complicated than the standard supervised settings (e.g., image classification) considered in Park et al. (2023). In particular, one challenge is that the diffusion model outputs a high-dimensional vector (an image) as opposed to a single scalar (e.g., a label). Even if we approximate the diffusion model as a *linear* model in parameters, naively applying TRAK would require keeping track of $p$ gradients for each training example (where $p$ is the number of pixels) and thus be computationally infeasible. Nonetheless, the presence of a single training example influences the optimization trajectory *only* via the gradient of the loss on that example—specifically, the MSE of the denoising objective. Hence, it suffices to keep track of a single gradient for each example. This observation allows us to estimate the change in model parameters using the same approach that TRAK uses (see Section 2). An additional challenge is that the gradient updates in the diffusion process are highly stochastic due to the sampling of random noise. To mitigate this stochasticity, we average the training loss over multiple resampling of the noise at randomly chosen steps and compute gradients over this averaged loss.

**A model output function for diffusion models.**    In Section 3, we motivated why we would like to attribute properties of the conditional distribution $p_{\theta(S)}(\cdot|\mathbf{x}_t)$, i.e., the distribution that arises from sampling when conditioning on an intermediate latent $\mathbf{x}_t$. Specifically, we would like to understand what training data causes the model to generate samples from this distribution. Then, one natural model output function $f_t$ would be to measure the likelihood of the model to generate these samples. Attributing with respect to such a choice of $f_t$ allows us to understand what training examples increase or decrease this likelihood.

In order to efficiently implement this model output function, we make two simplifications. First, sampling from $p_{\theta(S)}(\cdot|\mathbf{x}_t)$ can be computationally expensive, as this would involve repeatedly resampling parts of the diffusion trajectory. Specifically, sampling once from $p_{\theta(S)}(\cdot|\mathbf{x}_t)$ requires

applying the diffusion model $t$ times—in practice, $t$ can often be as large as 1000. Fortunately, as we described in Section 2, we can use the one-step estimate $\hat{\mathbf{x}}_0^t$ as a proxy for samples from $p_{\theta(S)}(\cdot|\mathbf{x}_t)$, since it approximates this distribution's expectation $\mathbb{E}_{\mathbf{x}_0 \sim p_\theta(\cdot|\mathbf{x}_t)}[\mathbf{x}_0]$. Second, it is computationally expensive to compute gradients with respect to the exact likelihood of generating an image. So, as a more tractable proxy for this likelihood, we measure the reconstruction loss (i.e., how well the diffusion model is able to denoise a noisy image) when adding noise to $\hat{\mathbf{x}}_0^t$ with magnitude matching the sampling process at step $t$. Specifically, we compute the Monte Carlo estimate

$$f_t\left(\hat{\mathbf{x}}_0^t, \theta(S)\right) = \sum_{i=1}^k \left\| \boldsymbol{\varepsilon}_i - \boldsymbol{\varepsilon}_{\theta(S)}\left(\sqrt{\bar{\alpha}_t}\hat{\mathbf{x}}_0^{(t)} + \sqrt{1-\bar{\alpha}_t}\boldsymbol{\varepsilon}_i, t\right)\right\|_2^2, \tag{3}$$

where $\bar{\alpha}_t$ is the DDPM[4] variance schedule (Ho et al., 2020), $\boldsymbol{\varepsilon}_i \sim \mathcal{N}(0,1)$ for all $i \in [k]$, and $k$ is the number of resampling rounds of the random noise $\boldsymbol{\varepsilon}$. Now that we have chosen our model output function, we can simply compute gradients with respect to this output to obtain the second component in Equation (1).

## 5 EXPERIMENTS

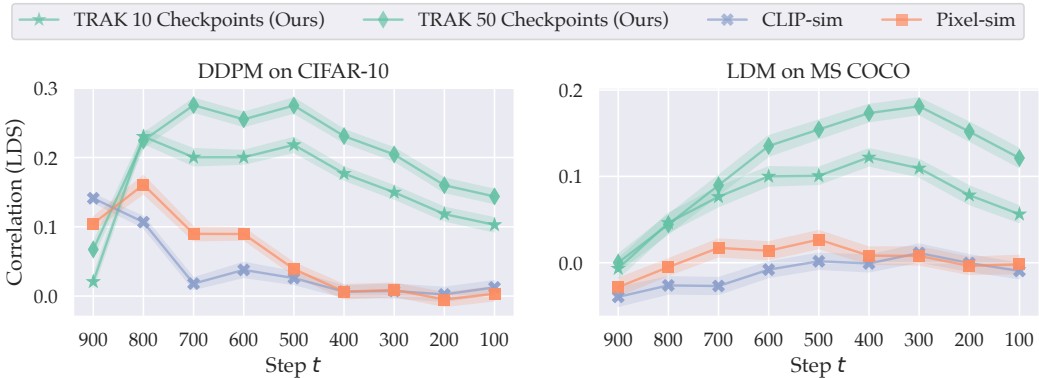

Figure 4: **Predicting model behavior.** We evaluate the counterfactual predictiveness of attributions using the LDS score at every 100 steps of the diffusion trajectory for TRAK (computed using 10 and 50 model checkpoints), as well as CLIP and pixel similarity. Smaller steps are closer to the final sample. Shaded areas represent standard error.

To evaluate our attribution method, we apply it to DDPMs trained on CIFAR-10 and LDMs trained on MS COCO We first visually interpret our attributions (Section 5.1), and evaluate their counterfactual significance (Section 5.2). Then, we explore how our attributions can be localized in pixel space (Section 5.3), as well as how they can be used to attribute the full diffusion trajectory (Section 5.4).

**Experimental Setup.** We compute our attribution scores using 100 DDPM checkpoints trained on CIFAR-10 and 50 LDM checkpoints trained MS COCO (See Appendix A for training details.). As baselines, we compare our attributions to two common image similarity metrics—CLIP similarity (i.e., cosine similarity of the CLIP embedding space) and cosine similarity in pixel space. We compute LDS scores at every 100 steps (of the 1000 steps in the DDPM scheduler) based on models trained on 100 random 50% subsets of CIFAR-10 and MS COCO. For our counterfactual evaluation in Section 5.2, we compute attribution scores on 50 samples from our CIFAR-10 and MS COCO models at step $t = 400$. Given the attribution scores for each sample, we then retrain the model after removing the corresponding top $k$ influencers for $k = 200, 500, 1000$. We sample 5000 images from two distributions: (1) the distribution arising from repeatedly initializing at $\mathbf{x}_{400}$ and sampling the final 400 steps from the original model; and (2) the distribution arising from repeating the above process but using the retrained model only for steps $t = 400$ to $t = 300$. We then compute FID distance between these distributions, and repeat this process for each sample at each value of $k$.

---

[4]We only consider DDPM schedulers in this work. The above derivation can be extended to other schedulers.

## 5.1 QUALITATIVE ANALYSIS OF ATTRIBUTIONS

In Figure 1, we visualize the sampling trajectory for an image generated by an MS COCO model, along with the most positive and negative influencers identified by TRAK (see Appendix C.5 for an equivalent visualization on CIFAR-10). We find that positive influencers tend to resemble the generated image throughout, while negative influencers tend to differ from the generated image along specific attributes (e.g., class, background, color) depending on the step. Interestingly, the negative influencers increasingly resemble the generated image towards the end of the diffusion trajectory. In Appendix C.3, we explore why negative influencers might reflect features of the final generated image, and conclude that once a feature of the final image is "decided," negative influencers will manifest this feature, as there is no possibility of "steering" the trajectory away from it. See Appendix C.5 for further examples of our attributions.

## 5.2 COUNTERFACTUALLY VALIDATING THE ATTRIBUTIONS

We now evaluate our attributions using the metrics introduced in Section 3.2 to validate their counterfactual significance. In Figure 4, we plot LDS scores for CIFAR-10 (left) and MS COCO (right) over a range of steps for our attribution scores as well as the two similarity baselines. Unlike in many computer vision settings (Zhang et al., 2018), we find that for CIFAR-10, similarity in pixel space achieves competitive performance, especially towards the start of the diffusion trajectory. However, for both CIFAR-10 and MS COCO, only TRAK is counterfactually predictive across the entire trajectory.

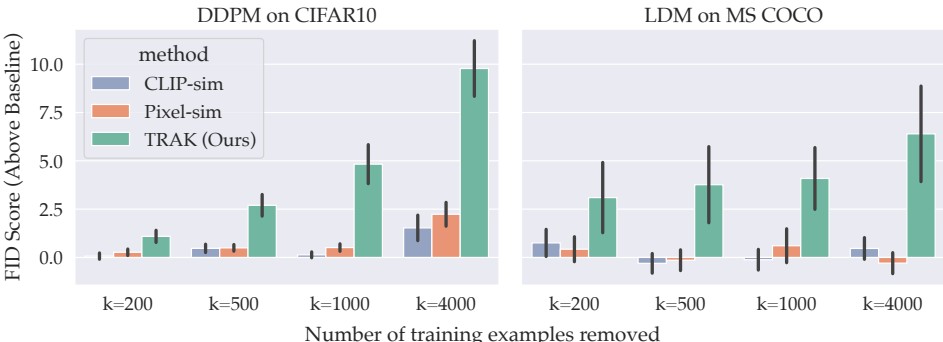

Figure 5: **Retraining without top influencers.** We plot the change to the distribution of generated images $p_\theta(\cdot|\mathbf{x}_{400})$ when substituting the original model with a new model only between steps $400$ and $300$. This new model is trained without the $k$ top influencers of $\mathbf{x}_{400}$ according to attributions from TRAK (computed at timestep $400$), CLIP similarity, and pixel similarity. We report the increase in FID score over a baseline of models trained on the full dataset. See Section 3.2 for details. Black bars represent standard error.

In Figure 5, we display the average FID scores (a measure of distance from the original model) after removing the $k$ most influential images for a given sample across possible values of $k$. Across values of $k$, removing the top influencers identified by our attribution method has a greater impact than removing the most similar images according to CLIP and pixel space similarities.

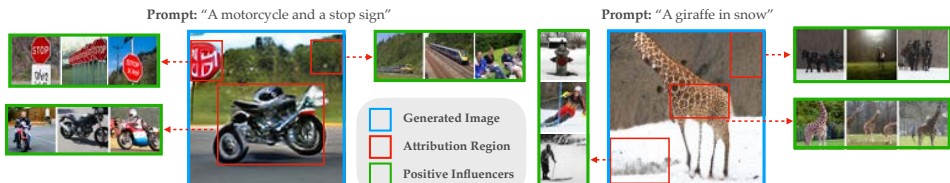

Figure 6: **Patch-based attribution.** We adapt our method to restrict attribution to user-specified patches of a generated image. We show examples of attributing patches capturing individual concepts in images synthesized by a latent diffusion model trained on MS COCO.

### 5.3 Localizing our attributions to patches in pixel space

Here we explore one possible direction towards attributing individual features: selecting a region of pixels (i.e., a *patch*) in a generated sample corresponding to a feature of interest, and restricting our model output function to this region. This way, we can restrict attributions only to the selected patch. To implement this model output function, we apply a pixel-wise binary mask to Equation (3) and ignore the output outside of the masked region. To test this approach, we generate images containing multiple features with an MS COCO-trained LDM. We then manually create per-feature masks for which we compute attribution scores with our method (see Figure 6). The resulting attributions for different masks surface training examples relevant *only* to the features in that region.

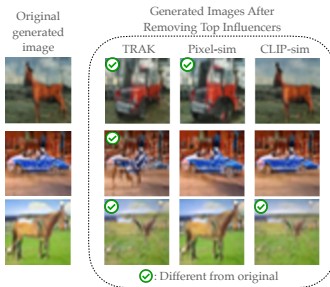 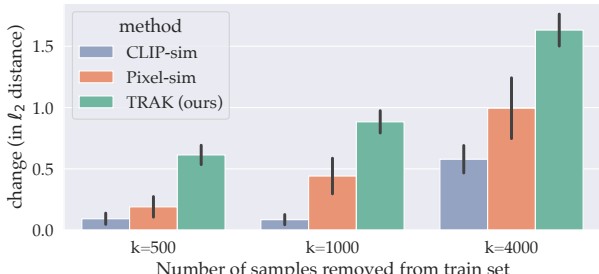

Figure 7: **"Forgetting" an image.** We quantify the impact of removing the highest scoring training examples according to TRAK, CLIP similarity, and pixel similarity. **(Left)** We compare the original synthesized samples to those generated from the same random seed with retrained models. **(Right)** To quantify the impact of removing these images, we measure the $\ell_2$ distance between 60 synthesized samples and corresponding images generated after retraining. Black bars represent standard error.

### 5.4 "Forgetting" how to generate an image

Our attribution scores and evaluation metrics are all step-specific. However, in practice we might care about identifying training images that impact the *full* diffusion pipeline. In particular, we might be interested in whether removing the important training images for a given synthesized image causes the diffusion model to "forget" how to generate this image. Specifically, given a set of attribution scores for a synthesized image, we remove the top $k$ influencers, retrain the model, and generate new images from scratch using the same random seed. Note that we leverage the fact that two diffusion models trained on the same dataset tend to generate similar images given the same random seed (see Appendix C.1 for more details). We then compare the change in pixel space between the original and newly generated image. This process is distinct from our evaluation metric, as (1) we directly compare two images rather than measure the distance between distributions, and (2) we re-generate images with our new model from scratch rather than restarting from some intermediate latent $\mathbf{x}_t$ and substituting the new model for only a small interval of steps (between $t$ and $t - \Delta$). We perform this process for our attribution scores on CIFAR-10 as well as the two similarity baselines (see Figure 7). Our results suggest that TRAK is able to identify influential images that have a significant impact on the full diffusion trajectory of the diffusion model.

## 6 Conclusion

In this work, we introduce a framework for data attribution for diffusion models and provide an efficient method for computing such attributions. In particular, we formalize data attribution in the diffusion setting as the task of quantifying the impact of individual training images on the generation of a given image *at each step* of the diffusion process. We additionally provide two metrics for evaluating such attributions, and apply these metrics to validate our attributions for DDPMs trained on CIFAR-10 and LDMs trained on MS COCO.

## REPRODUCIBILITY STATEMENT

We provide our code at https://anonymous.4open.science/r/iclr-diffusion-code-0AAB. The code uses PyTorch (Paszke et al., 2019) and is a lightweight wrapper around the TRAK library provided by Park et al. (2023) (https://github.com/MadryLab/trak). Our code can be used to reproduce all necessary components to compute our attribution scores as described in Section 4. For training all models in our paper, we use off-the-shelf code from the diffusers library (https://huggingface.co/docs/diffusers/v0.13.0/en/training/text2image).

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

# Appendices

# A    EXPERIMENTAL DETAILS

Throughout our paper, we train various diffusion models on CIFAR-10 and MS COCO.

**DDPM training on CIFAR-10.**    We train 100 DDPMs (Ho et al., 2020) on the CIFAR-10 dataset for 200 epochs using a cosine annealing learning rate schedule that starts at 1e-4. We used the DDPM architecture that match the original implementation (Ho et al., 2020), which can be found here `https://huggingface.co/google/ddpm-cifar10-32`. At inference time we sample using a DDPM scheduler with 50 inference steps.

**LDM training on MS COCO.**    We train 20 text-conditional latent diffusion models (LDMs) (Rombach et al., 2022) on the MS COCO dataset for 200 epochs using a cosine annealing learning rate schedule that ståarts at 2e-4. We use the exact CLIP and VAE used in Stable Diffusion v2, but use a custom (smaller) UNet, which we describe in our code. These models can be found here `https://huggingface.co/stabilityai/stable-diffusion-2-1`. At inference time, we sample using a DDPM scheduler with 1000 inference steps.

**LDS.**    We sample 100 random $50\%$ subsets of CIFAR-10 and MS COCO, and train 5 models per mask. Given a set of attribution scores, we then compute the Spearman rank correlation (Spearman, 1904) between the predicted model outputs $g_\tau(\cdot)$ (see Eq. (2)) on each subset according to the attributions and the (averaged) actual model outputs. Because our model output and attributions are specific to a step, we compute LDS separately for each step. To evaluate the counterfactual significance of our attributions over the course of the diffusion trajectory, we measure LDS scores at each 100 steps over the 1000 step sampling process.

**Retraining without the most influential images.**    For our counterfactual evaluation in Section 5.2, we compute attribution scores on 50 samples from our CIFAR-10 and MS COCO models at step $t = 400$. Given the attribution scores for each sample, we then retrain the model after removing the corresponding top $k$ influencers for $k = 200, 500, 1000$. We compute FID based on 5000 images from each distribution, and repeat this process for each sample at each value of $k$.

**TRAK hyperparameters**    In the random projection step of TRAK, we use a projection dimension of $d = 4096$ for CIFAR-10 and $d = 16384$ for MS COCO. As in Park et al. (2023), we use multiple model checkpoints in order to compute the attribution scores. For CIFAR-10, we use 100 checkpoints, and for MS COCO, we use 20 checkpoints. After the de-anonymization of the paper, we will release the pre-computed TRAK features for all of our models, allowing for a quick computation of TRAK scores on new synthesized images. In Equation (3), we use $k = 20$ for both CIFAR-10 and MS COCO.

## B    RELATED WORK

**Data Attribution.**    A long line of work has studied the problem of training data attribution, or tracing model behavior back to training data; we focus here on works done in the context of modern machine learning algorithms. Prior approaches include those based on the influence function and its variants (Hampel et al., 2011; Wojnowicz et al., 2016; Koh & Liang, 2017; Basu et al., 2019; Khanna et al., 2019; Achille et al., 2021; Schioppa et al., 2022; Bae et al., 2022), sampling-based methods that leverage models trained on different subsets of data (Ghorbani & Zou, 2019; Jia et al., 2019; Feldman & Zhang, 2020; Ilyas et al., 2022; Lin et al., 2022), and various other heuristic approaches (Yeh et al., 2018; Pruthi et al., 2020). These methods generally exhibit a strong tradeoff between predictiveness or effectiveness and computational efficiency Jia et al. (2021). The recent method of Park et al. (2023) significantly improves upon these tradeoffs by leveraging the empirical kernel structure of differentiable models. While most prior work primarily focus on the supervised setting, more recent works study attribution in generative settings, including to language models (Park et al., 2023) and to diffusion models (Wang et al., 2023).

**Memorization in Generative Models.**    Prior to the increasing popularity of diffusion models, a number of previous works studied memorization in other generative models. For example, Feng et al. (2021) study the impact of properties of a dataset (size, complexity) on training data replication in Generative Adversarial Networks (GANs), and van den Burg & Williams (2021) introduce a memorization score for Variational Autoencoders (VAEs) that can be additionally applied to arbitrary generative models. Following the release of large text-to-image diffusion models, the creators of one of these models (DALL·E 2) investigated memorization issues themselves and found that memorization could be significantly decreased through de-duplication of the training data (Nichol et al., 2022). Recently, Somepalli et al. (2022) explored the data replication behavior of diffusion models from the lens of 'digital forgery,' and identified many cases where, even when Stable Diffusion produced 'unique' images, it directly copies style and semantic structure from individual images in the training set. On the other hand, Carlini et al. (2023) investigate memorization from the perspective of privacy, and show that query access to diffusion models can enable an adversary to directly extract the models' training data.

# C ADDITIONAL ANALYSIS AND RESULTS

## C.1 DIFFUSION MODELS ARE CONSISTENT ACROSS SEEDS

A priori, two independent models trained on the same dataset do not share the same latent space. That is, a given noise sequence $\varepsilon_T, ..., \varepsilon_0$ could be denoised to two unrelated images for two different models. However, we find empirically that latent spaces from two diffusion models are highly aligned; we call this property *seed consistency*. In fact, we find that images generated by many independently trained DDPMs on CIFAR-10 from the same random seed and nearly indistinguishable (see Figure C.1, right). To evaluate seed consistency quantitatively, we measure the $\ell_2$ distance between images generated by two models when using identical or distinct noise sequences, and find that matching the noise sequences leads to a far smaller $\ell_2$ distances (see Figure C.1, left).

We additionally evaluate seed consistency on multiple checkpoints of Stable Diffusion (we use checkpoints provided at https://huggingface.co/CompVis/stable-diffusion and https://huggingface.co/runwayml/stable-diffusion-v1-5. and find that images generated across these models with a fixed seed share significantly more visual similarity that those generated from independent random seeds (see Figure C.2.)

We take advantage of this property when evaluating the counterfactual impact of removing the training examples relevant to a given generated image (see Section 5.4). Specifically, we now expect that retraining a model on the full training set and then sampling from the same seed should produce a highly similar image to the generated image of interest. Thus, we can evaluate the counterfactual significance of removing the training examples with the top attribution scores for a given generated image by retraining and measuring the distance (in pixel space) of an image synthesized with the same seed to the original generated image.

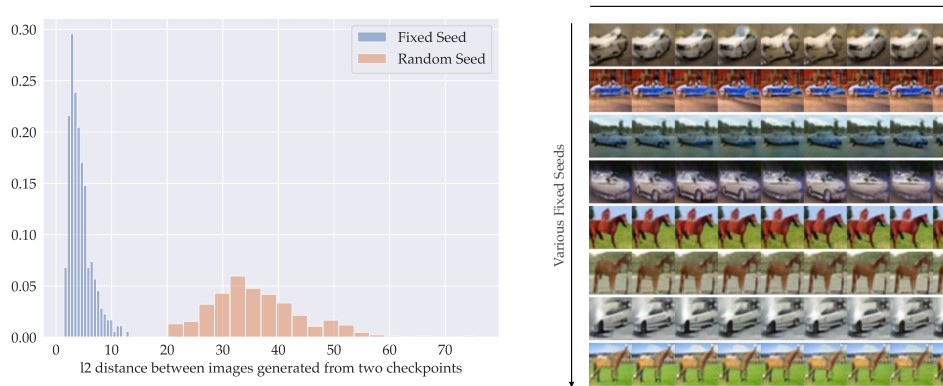

Figure C.1: **Seed consistency of CIFAR-10 DDPMs**. We find that across DDPMs trained independently on CIFAR-10, when using a fixed random seed during sampling, the resulting synthesized images are very similar, and often visually indistinguishable **(Right)**. Quantitatively, we find that the $\ell_2$ distance between images generated from two different models is significantly smaller when we fix the noise sequence **(Left)**.

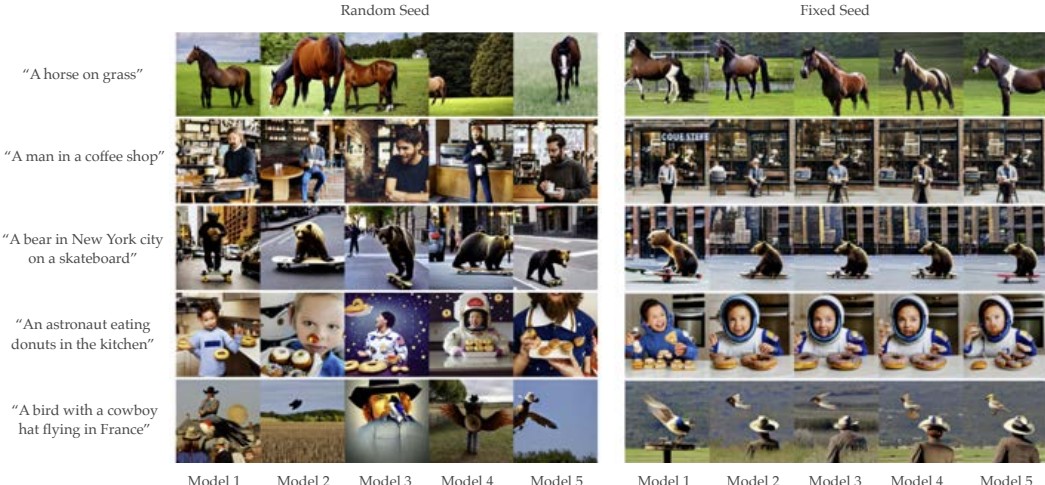

Figure C.2: **Seed consistency holds for Stable Diffusion models.** We find that seed consistency holds even for large, text conditioned model, specifically for Stable Diffusion models that are trained on LAION-5B. We compare multiple checkpoints of Stable Diffusion provided by Stability AI, and find that fixing the noise sequence during sampling surfaces very similar images (in comparison to using independent noising sequences).

## C.2 ATTRIBUTION SCORES CAN DRASTICALLY CHANGE OVER THE COURSE OF THE DIFFUSION PROCESS

As additional motivation for performing attribution at individual steps rather than the entire diffusion trajectory, we highlight the following phenomena: *the same training image can be both positively influential and negatively influential for a generated sample at different steps*. For example, consider an image of a red car on a grey background generated by our DDPM trained on CIFAR-10 (See Figure C.3, top). We find that a specific training example of a red car on grass is the single most positively influential image according to TRAK at the early stages of the generative process (as it is forming the shape of the car), but is later the single most negatively influential image (possibly due to the difference in background, which could steer the model in a different direction). If we were to create an aggregate attribution score for the entire diffusion trajectory, it is unclear what the attribution score would signify for this training example.

To evaluate this phenomena quantitatively, we measure the percentage of generated images for which, for a given $K$, there exists a training example that is one of the top $K$ highest scoring images at some step and one of the top $K$ lowest scoring images at another step (according to TRAK). In Figure C.4, we show how this percentage varies with $K$. As a baseline, we also include the probability of such a training example existing given completely random attribution scores. We find that our observed probabilities match those expected with random scores, signifying that an image being highly positively influential at a given step *does not* decrease the probability that it is highly negatively influential at a different step.

To more broadly analyze the relationship between attributions at different steps, we additionally measure the Spearman's rank correlation (Spearman, 1904) between attribution scores for the same generated sample at different steps (see Figure C.5). We find that for steps that are sufficiently far from each other (around 500 steps), the attribution scores are nearly uncorrelated.

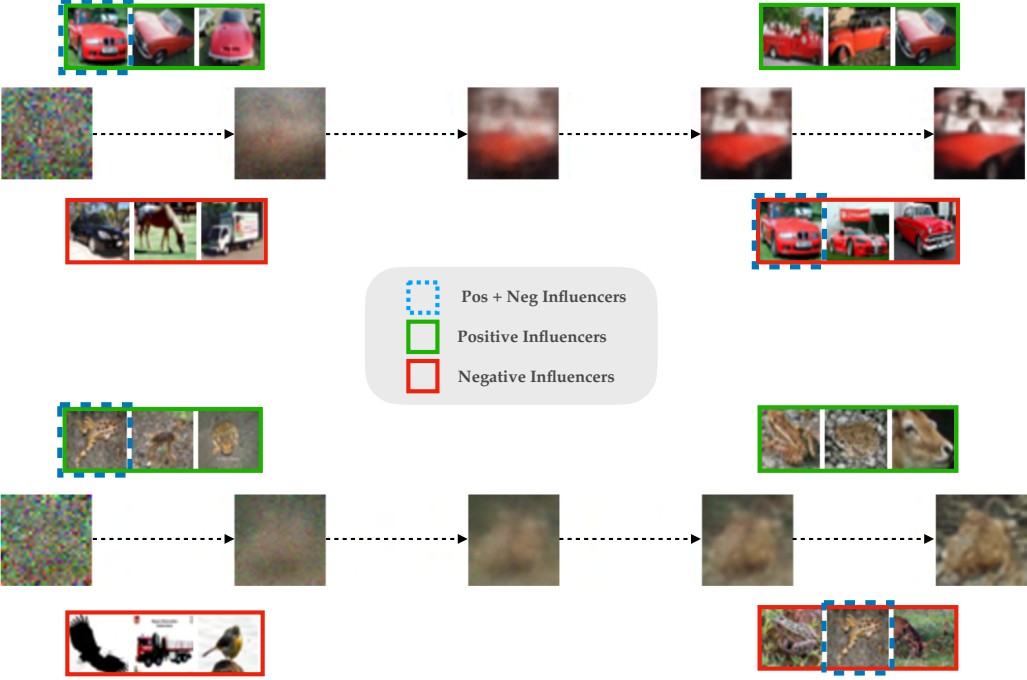

Figure C.3: **Overlap between positive and negative influencers.** Here, we visualize the generative process for two images generated by a DDPM on CIFAR for which there exists a training image that is both positively and negatively influential at different steps. If we consider an aggregate attribution score across all time-steps of the diffusion trajectory, we might lose the significance of such training examples which alternate between being positively and negatively influential during the sampling process.

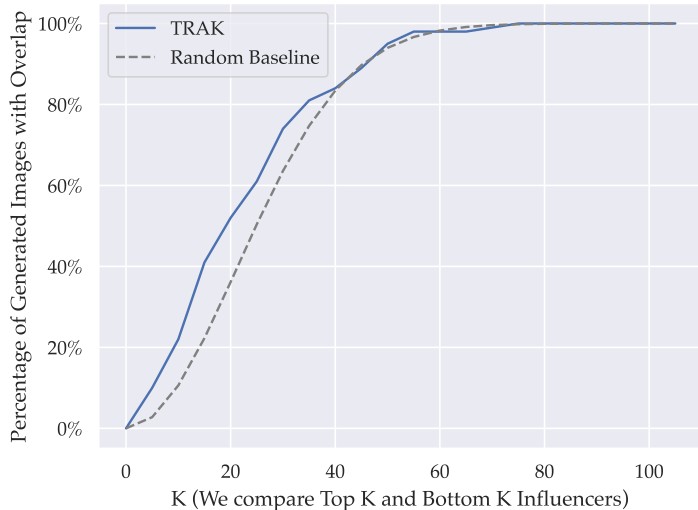

Figure C.4: **The relationship between positive and negative influencers.** Here, we plot the probability that within the attribution scores for a given generated image, there exists a training example that is one of the $K$ most positive influencers at some step and one of the bottom $K$ most negative influencers at another step. We compute this probability empirically with the attribution scores from TRAK and find that it closely aligns with the hypothetical baseline of completely random attribution scores. This signifies that being a top positive influencer at some step does not decrease the likelihood of being a top negative influencer at a different step.

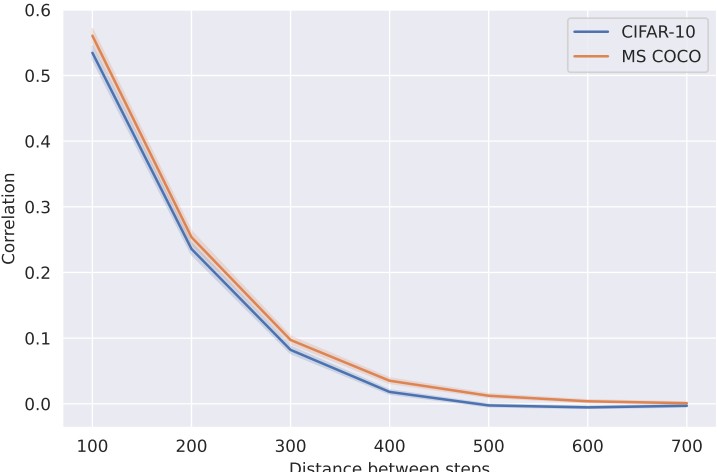

Figure C.5: **Correlation between attribution scores over Steps.** Here, we plot the Spearman's rank correlation (Spearman, 1904) between the attribution scores for a given image generated by either our CIFAR-10 or MS COCO models at different steps, as a function of the distance between steps (results are averaged over 100 generated samples). As expected, steps that are closer in proximity have more closely aligned attribution scores. Interestingly, when we compute attributions at steps of distance 500 or more apart, the resulting scores are nearly uncorrelated.

### C.3 WHY DO NEGATIVE INFLUENCERS RESEMBLE THE FINAL GENERATED IMAGE?

To answer this question, we study the relationship between the top (positive and negative) influencers and the distribution $p_\theta(\cdot|\mathbf{x}_t)$ towards which we target our attributions. Specifically, for a given generated image $\mathbf{x}_0$ and feature of interest, which we can represent with a binary function $f : \mathcal{Z} \to \{0, 1\}$, we can measure $\mathbb{E}[f(p_\theta(\cdot|\mathbf{x}_t))]$, i.e., the probability of observing the feature $f$, as a function of $t$; now, we can identify how our attributions relate to changes in $f$.

In Figure C.6, for a given synthesized image of a horse by our CIFAR-10 model, we plot the likelihood that images from $p_\theta(\cdot|\mathbf{x}_t)$ containing a horse (according to a CIFAR-10-trained classifier) as a function of the step $t$ (left), along with the top and bottom influencers at three points along the trajectory (right). We find that the top influencers begin to reflect the feature of interest once the likelihood of this feature begins to grow, yet once the likelihood of the feature reaches near certain, the negative influencers *also* begin to reflect this feature. Indeed, this makes intuitive sense, as after this point it would be impossible to "steer" the trajectory away from presenting this feature. So, the negative influencers at these later steps might now steer the trajectory away from another of its final features (e.g., the color of horse) that has not yet been decided at that step.

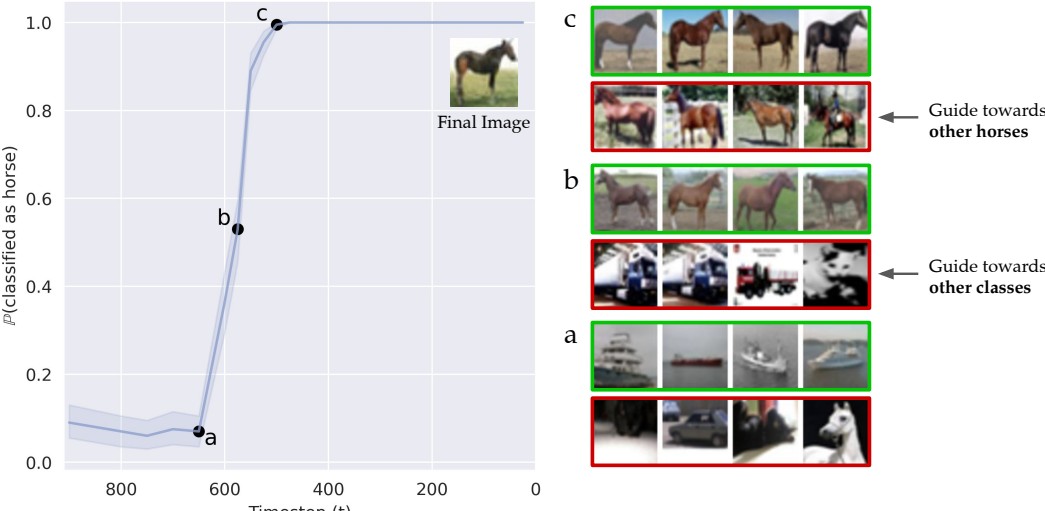

Figure C.6: **Relating Attributions to the Diffusion Trajectory.** (**Left**) For a given image of a horse generated by a CIFAR-10 DDPM model, we plot the likelihood that samples from the distribution $p_\theta(\cdot|\mathbf{x}_t)$ (see Section 2) are classified as a horse according to a CIFAR-10 model. This likelihood increases rapidly around steps 650 to 500. (**Right**) At three steps within this range, we visualize the most influential training examples (positive in green, negative in red) identified by our method. Note that once the horse feature is "decided" around step 500, the negative influencers begin to reflect this feature.

### C.4 FEATURE ANALYSIS FOR STABLE DIFFUSION

We analyze how the likelihood of different features in the final image varies over steps for images generated by a Stable Diffusion model,[5] similarly as we did for CIFAR-10 in Figure 3. To measure the relative likelihood between two features (e.g., "white blouse" vs. "dark blouse"), we use a pretrained CLIP model and measure whether the CLIP embedding of the generated image is closer to the text embedding of the first feature or the second feature. We sample 60 images at each timestep and report the average likelihood. We use 300 denoising steps to speed up the generation.

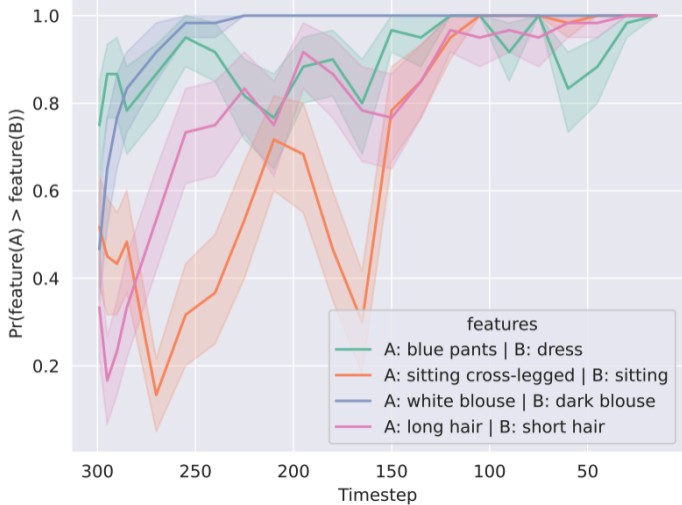
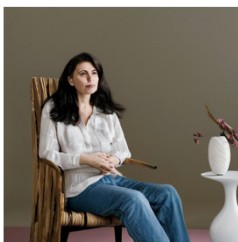

target image

Figure C.7: **Features appear at specific steps for Stable Diffusion. (Left)** For each pair of features, we plot the evolution in the relative likelihood of the two features (according to CLIP text-image similarity) in the conditional distribution $p_\theta(\cdot|\mathbf{x}_t)$. Features differ in when they appear, but usually rapidly appear within a short time interval. (**Right**) The generated image $\mathbf{x}_0$, sampled using $T = 300$ denoising steps.

---

[5]We use the `stabilityai/stable-diffusion-2` pretrained checkpoint.

## C.5 OMITTED PLOTS

In this section, we present additional visualizations extending upon the figures in the main text. In Figure C.8 and Figure C.9, we visualize the most influential training examples identified by our method for a sample generated with a DDPM trained on CIFAR-10 and a LDM trained on MS COCO, respectively. In Figure C.10, we more concisely display attributions for additional samples generated by a CIFAR-10 DDPM. Finally, in Figure C.11 we display additional examples of the appearance of features over steps, and confirm that our findings in the main text hold across when different classification models are used for identifying a given feature.

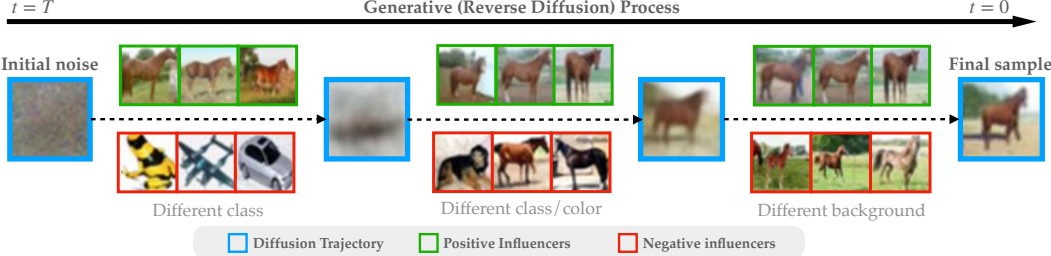

Figure C.8: An example of time-dependent attribution scores for a sample generated by a DDPM trained on CIFAR-10. At each step $t$, our method pinpoints the training examples with the highest influence (positive in green, negative in red) on the generative process at this step. In particular, positive influencers guide the trajectory towards the final sample, while negative influencers guide the trajectory away from it.

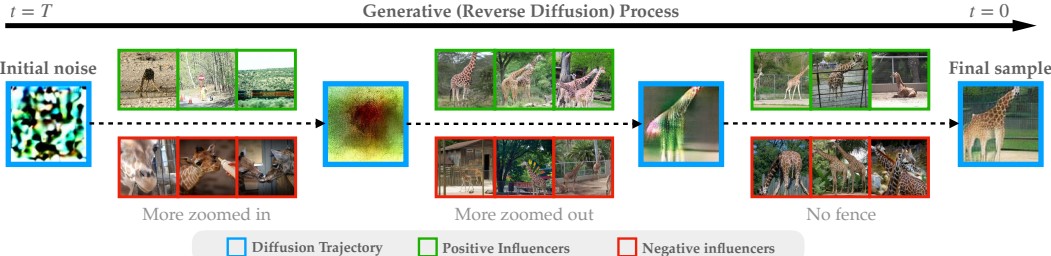

Figure C.9: An additional example of time-dependent attribution scores for a sample generated by a LDM trained on MS COCO. At each step $t$, our method pinpoints the training examples with the highest influence (positive in green, negative in red) on the generative process at this step. In particular, positive influencers guide the trajectory towards the final sample, while negative influencers guide the trajectory away from it.

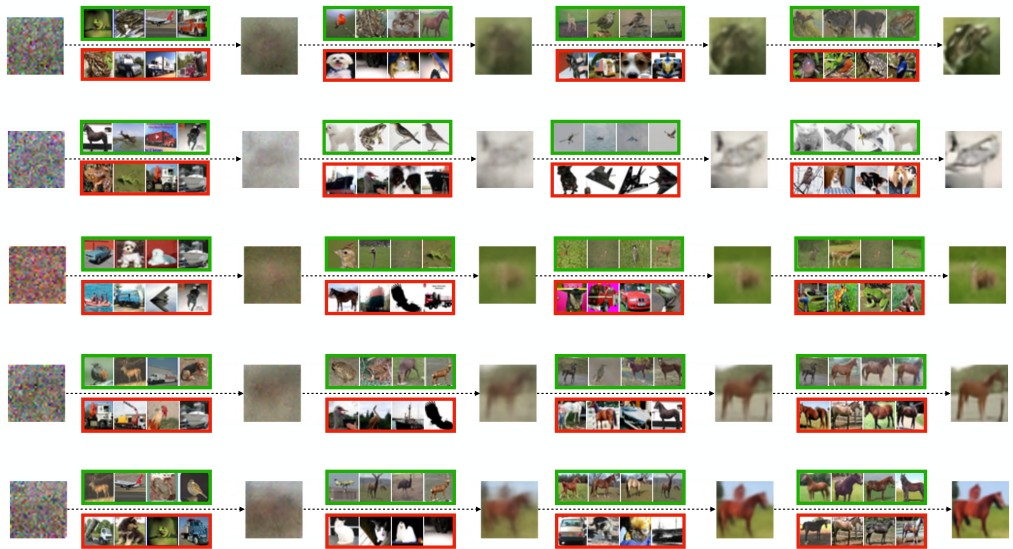

Figure C.10: Additional examples our attributions identified by our method. Here, we visualize the diffusion trajectory for generated images along with the most positively (green) and negatively (red) influential images at individual steps throughout the diffusion trajectory.

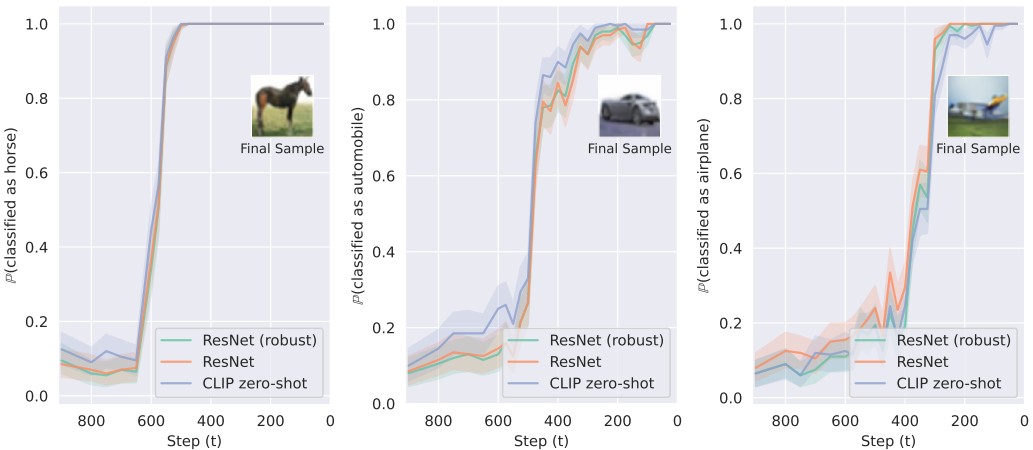

Figure C.11: Additional examples of the appearance of features over steps, similar to the analysis in Figure 3. In each plot, we show the likelihood that a sample generated from the distribution $p_\theta(\cdot|\mathbf{x}_t)$ contains a the feature of interest (in this case, the CIFAR-10 class of the final image) according to three different classifiers: a ResNet trained on the CIFAR-10 dataset with either standard or robust training, and zero-shot CLIP-H/14 model (Radford et al., 2021). Note that in each example, the likelihood that the final image contains the given feature increases rapidly in a short interval of steps, and that this phenomena is consistent across different classifiers.

