# OpenReview forum: "The Journey, Not the Destination: How Data Guides Diffusion Models"
_ICLR.cc/2024/Conference — Submitted to ICLR 2024_

### Official Review · Reviewer_ujFU · 2023-10-25

**Soundness:** 3 good
**Presentation:** 2 fair
**Contribution:** 3 good
**Rating:** 6
**Confidence:** 4

**Summary:**

The authors propose a framework for data attribution in diffusion models to attribute generated images back to the training data, allowing for identification of influential training examples. The framework attributes each step of the diffusion process, providing targeted attributions for specific features of the final generated image. The paper introduces metrics for evaluating the attributions and presents a method for computing them.

**Strengths:**

Strengths:
1) The paper clearly conveys the significance of data attribution in diffusion models, the theoretical validation of the framework, and the development of metrics for evaluating attributions.
2) The paper also demonstrates the practical applicability of the framework through experiments on real datasets.
3) Detailed implementation settings and code are provided, which makes for easy reproducing the study.

**Weaknesses:**

Weaknesses:
1) The structure of the paper is a bit confusing and not easy to follow.
2) The paper formulated a novel research problem, but the proposed method appears to be relatively primitive compared to the complexity of the problem at hand.

**Questions:**

Please refer to the weakness part. Further, the fine-grained analysis in section C.2 is intriguing. It appears that the model has a tendency to generate images by referencing specific semantic parts of the training images rather than generating the entire images. Therefore, does it still make sense to attribute these generated images to image-level data?

---

> ### Author Response · Authors · 2023-11-16
> **Rebuttal Response**
>
> We thank the reviewer for the helpful feedback.
>
>
> ### Structure of the Presentation
> To address the reviewer’s concerns over the structure of the presentation, we have made a number of changes to the structure of the main text:
> We made revisions to Section 3 (which introduces our attribution framework) to simplify the language and make the presentation more modular. Additionally, we added an in-depth section on TRAK [1] to the background section in order to better contextualize our method.
> We shortened Section 4 (which describes our method) and made more explicit connections to the preliminaries in Section 2.
>
> ### Simplicity of the proposed method
> As the reviewer acknowledges, our paper formalizes and studies a new research problem. We introduce a method building on top of state-of-the-art work for data attribution in the supervised setting. While our method is relatively simple, we view the simplicity of our method as a feature, not a bug. In particular, we demonstrate through our evaluations that, despite its “simplicity,” our method surfaces attributions that are counterfactually significant to the behavior of diffusion models. Due to its simplicity, our method is easier to implement (by extending upon the open source implementation of TRAK) and easier for other works to build upon. Conceptually, the success of our approach indicates that despite the complexity underlying modern NNs, we can reason about them by approximating them with kernel models.
>
> ### Attribution beyond image-level data
> We agree that it would be interesting to extend the data attribution task to attribute behavior not only to individual training examples, but rather to parts of images (our understanding is that is what the reviewer is suggesting). For example, the diffusion model might be using the background of some training image at an earlier step, but might then use an object in the foreground at a later point (we allude exactly to this in Appendix C.2). More broadly, such a perspective on data attribution  would also be useful in other settings, including supervised tasks. We view this direction as a great potential future extension of the work.
>
> However, we believe that it is  still very valuable to attribute generated samples back to image-level data. First, for many possible applications of data attribution, it is most convenient and natural to intervene on image-level data. For example:
> Copyright violations and data leakage are generally only discussed at the level of individual images (or sets of images, e.g., an artist’s body of work).
> Data curation is almost always formalized on a per-image level.
> Data poisoning is usually performed on individual examples [2].
>
> Additionally, attributing with respect to patches might limit the scalability of the method. For example, if we divide images into 16 patches, we would need to compute and store 16x more gradients.
>
> [1] Park, S. M., Georgiev, K., Ilyas, A., Leclerc, G., & Madry, A. (2023). TRAK: Attributing model behavior at scale. arXiv preprint arXiv:2303.14186.
>
> [2] Khaddaj, Alaa, et al. "Rethinking Backdoor Attacks." (2023).

---

> > ### Author Response · Authors · 2023-11-22
> > **Reminder**
> >
> > As the author-reviewer discussion period comes to a close, we'd like to ask if your comments have been sufficiently addressed by the rebuttal. We'd be happy to continue the discussion if you have further questions.
> >
> > Thank you again for your review!

---

### Official Review · Reviewer_XavF · 2023-10-31

**Soundness:** 3 good
**Presentation:** 1 poor
**Contribution:** 3 good
**Rating:** 6
**Confidence:** 3

**Summary:**

In this work, authors introduce an extension of the Trek method that provides data attribution measurement to the family of diffusion methods. The goal of the analysis is to show that with the proposed methodology we can attribute which training images (and how) influenced the final generation. The evaluation is performed on CIFAR10 and MS COCO dataset with DDPMs and latent diffusion models.

**Strengths:**

- This submission tackles an interesting and important problem that to my knowledge was not yet approached in the context of diffusion models
- The proposed method is a direct application of the Trek method to diffusion models. However, this extension is non-trivial and might have significant impact on some areas of research such as machine unlearning.
- The evaluation is performed on a big scale (although it is mostly presented in the appendices)

**Weaknesses:**

-In general, there is a significant mismatch between the goal of the method as expressed at the beginning of the submission and the final experiments being evaluated. This is due to a series of approximations that make the computation possible. The interesting question is how those approximations influence the final observations (e.g. using one-step approximation of $\hat{x}_0^t$ as an approximation of the distribution’s expectation $E[x_0|x_t]$). If I understand it correctly, this assumption indicates that there would be no change in the final distribution due to later diffusion steps. For some cases with higher t values (e.g. timesteps>600 in Figure 3) this approximation might be wrong.
- “However, the effect of a single time step on the final distribution may be small and hard to quantify. Hence, we assume that attributions change gradually over time and replace the denoising model for a small interval of time steps (i.e., between steps t and t − Δ).” - This assumption is counterintuitive to the previous derivations on what authors consider attributions for diffusion models in their framework.
- The submission is hard to follow without in-depth understanding of the Trak method (Perk et al. 2023) it is heavily based on. It is also almost impossible to understand the experiments without reading the appendix.
- The presentation of the submission seems ad hoc and sloppy. There are multiple different thoughts that lack cohesion. For example Figure 4 appears on page 5 without any connection to the text, while it is referred to on page 8.
- The provided code in its current form is far from being useful for the full reproduction of the results presented in this work. It is a total of ~200 useful lines of code (excluding imports and  comments) with some definitions of functions that should be connected to something in an unclear way.
- The submission does not follow ICLR guidelines - the margins are significantly smaller (1 in instead of required 1.5) and the font is different.

**Questions:**

- Are the CIFAR10 models evaluated in this submission conditioned on the class identity? Otherwise, how for example the plot in Fig 3 (left) calculated - using only 15 examples presented on the right?
- “Thus, if we treat this likelihood as a function of t, the steps at which there is the largest increase in likelihood (i.e., the steepest slope) are most responsible for the presence of this feature in the final image. In fact, it turns out that this likelihood often increases rapidly” -  The example presented in Fig 3 is rather extreme. Do you have any intuition why it is so?

**Details Of Ethics Concerns:**

The submission does not follow ICLR guidelines - the margins are significantly smaller (1 in instead of required 1.5) and the font is different.

---

> ### Author Response · Authors · 2023-11-16
> **Rebuttal Response (1/2)**
>
> We thank the reviewer for the helpful feedback.
>
> ### Mismatch between goal and final experiments
> The reviewer raises a concern that “there is a significant mismatch between the goal of the method as expressed at the beginning of the submission and the final experiments being evaluated […] due to a series of approximations”. We would like to respectfully disagree with this statement.
>
> On one hand, we acknowledge that our method indeed involves a series of approximations, which enable us to run large scale experiments. On the other hand, importantly, we do not make these simplifications in our evaluation. In particular, note that in Figure 5 (Retraining without top influencers), we simply retrain the diffusion models (DDPMs and LDMs) without the training examples we flag as influential, and sample directly from $p_\theta(\cdot | x_t)$ for each of the resulting models. Then, we report FID between those samples, and the ones generated by models trained on the entire training set. Note that neither of the approximations we employ in our method (using $\hat{x}\_0^t$ in lieu of samples from $p_\theta(\cdot|x_t)$, and using the training loss instead of the likelihood) is present in this evaluation. Hence, despite the approximations used in computing attribution scores, we show that our method significantly outperforms the similarity-based baselines in attributing the full distribution.
>
> The reviewer also raises the question whether replacing $\mathbb{E}\_{x\_0 \sim p\_{\theta}(\cdot|x\_t)}[x\_0]$ with $\hat{x}\_0^t$ implicitly creates an additional assumption that the later diffusion steps do not contribute to the final generated image. This is not the case: the use of $\hat{x}\_0^t$ to approximate $\mathbb{E}\_{x\_0 \sim p\_{\theta}(\cdot|x\_t)}[x\_0]$ simply provides us with an computationally-efficient estimate of $\mathbb{E}\_{x\_0 \sim p\_{\theta}(\cdot|x\_t)}[x\_0]$. As we note in the paper, the additional assumption we introduce is that the diffusion model is approximately consistent, as defined in [2].
>
> Overall, we found that despite the simplicity of the approximations used, our approach enables meaningful attributions. That said, using more sophisticated approximations is an interesting direction for future work.
>
> ### Effect of a single step
> The reviewer raises a concern about the assumption that our attributions change gradually over the course of the diffusion process. First, we note that we empirically observe this phenomenon. In particular, see Figure C.5---we show that there is a very high rank correlation between our attributions for steps in close proximity to each other. Second, we would like to point out that in practice we only use this assumption for small intervals, e.g., <=100 steps within a 1000-step diffusion trajectory.
>
> ### Presentation of the submission
> To address the reviewer’s concerns over the quality of the presentation, we have made a number of improvements to the structure of the main text:
> - “The submission is hard to follow without in-depth understanding of the Trak method”: We added an in-depth section on TRAK [1] to the background section in order to better contextualize our method.
> - “Figure 4 appears on page 5 without any connection to the text, while it is referred to on page 8”: Some of the figures were indeed poorly placed - we have adjusted the figure placements in the rebuttal, so that they appear closer to where they are referenced.
> - “lack of cohesion”: we made revisions to Sections 3 and 4 that introduce our attribution framework and describe our method. In particular, we simplified the language and made the presentation more modular.
>
>
> ### Code
> Indeed, we do provide a concise (~200 loc) API built on top of the TRAK API (https://github.com/madrylab/trak/). But we view the simplicity of our implementation as one of the strengths of our method rather than a shortcoming. To address the ease of use of our API, we have added:
> 1. Two demo notebooks (`examples/demo_MSCOCO.ipynb` and `examples/demo_CIFAR10.ipynb`) that provides a simple way for users to score images synthesized by an LDM model trained on MS COCO, and a DDPM model trained on CIFAR-10, respectively.
> 2. A set of scripts (together with a README with an explicit set of commands) to reproduce the computation of our TRAK scores on both DDPM models trained on CIFAR-10 and LDMs trained on MS COCO.
>
> ### Adhering to margin and font
> We thank the reviewer for pointing out the incorrect margins and font size in our submission. This was an unintentional accident due to a git merge, and we fixed the revised version to adhere to the ICLR guidelines. The margins we used were wider than the ICLR guidelines, however it turns out our font was also wider. After fixing both the font and margins, we found that there was very little difference in the resulting paper length (about one paragraph). We apologize for this error.

---

> > ### Author Response · Authors · 2023-11-16
> > **Rebuttal Response (2/2)**
> >
> > ### Calculation of Likelihood in Figure 3
> > Our CIFAR-10 DDPM models are unconditional. To calculate the likelihood of a given image containing a horse in Figure 3, we apply a CIFAR-10-trained ResNet classifier to 200 samples from the distribution $p_{\theta}(\cdot|x_t)$. 15 of these 200 samples are visualized in the figure. Following the comment of reviewer KCCM, we ablate this result over three different choices of classifiers for identifying the horse features and find that the plot is consistent in each case (see the newly added Figure C.11 in Appendix C.5).
> >
> >
> > ### Intuition for rapid increase in likelihood
> > “The example presented in Fig 3 is rather extreme. Do you have any intuition why it is so?”
> > Indeed, we note that the transition in Figure 3 happens in a remarkably short interval of steps. We acknowledge that the nature of our finding is empirical and we currently lack a theoretical understanding. One possible (very crude) intuition for the rapid increase in likelihood is that before the transition, the noisy latent $x_t$ is approximately equidistant to multiple “clusters” of images (e.g., a cluster of “horse” images and a cluster of “car” images), and after the transition $x_t$ is closer to one of these clusters, thus making the diffusion model “commit” to that cluster. Importantly, this would happen if the latent space in which the diffusion process happens is aligned with human-aligned features---i.e., images closer in the latent space are also semantically more similar to us. As some additional motivation for the existence of this phenomenon, we would like to note that discrete events like bifurcations often appear in continuous systems.
> >
> > [1] Park, S. M., Georgiev, K., Ilyas, A., Leclerc, G., & Madry, A. (2023). TRAK: Attributing model behavior at scale. arXiv preprint arXiv:2303.14186.
> >
> > [2] Daras, G., Dagan, Y., Dimakis, A. G., & Daskalakis, C. (2023). Consistent diffusion models: Mitigating sampling drift by learning to be consistent. arXiv preprint arXiv:2302.09057.

---

> > > ### Author Response · Authors · 2023-11-22
> > > **Reminder**
> > >
> > > As the author-reviewer discussion period comes to a close, we'd like to ask if your comments have been sufficiently addressed by the rebuttal. We'd be happy to continue the discussion if you have further questions.
> > >
> > > Thank you again for your review!

---

### Official Review · Reviewer_KmFt · 2023-11-01

**Soundness:** 2 fair
**Presentation:** 3 good
**Contribution:** 3 good
**Rating:** 6
**Confidence:** 4

**Summary:**

This work focuses on attribution for diffusion models, i.e. understanding how the underlying training data influences the generation of a sample. This work proposes an approach based on TRAK [1] to attribute a single denoising step back to the training data. The work also proposes metrics based on counterfactual estimation to evaluate attribution approaches and shows comparisons against simple baselines on multiple datasets (CIFAR-10 and MS-COCO).

[1] Park, Sung Min, et al. "Trak: Attributing model behavior at scale." _arXiv preprint arXiv:2303.14186_ (2023).

**Strengths:**

1. The work focuses on an important and timely problem. Attribution for models is an important technical problem to address as generative models become more ubiquitous. This has implications for regulation, copyright, and fair compensation to artists [1].
2. The proposed solution is reasonably motivated and builds on prior work that achieves SOTA attribution results for discriminative models. The work also compares against reasonable baselines for data attribution and shows results across two different datasets namely CIFAR-10 and MS COCO.
3. The proposed attribution results look reasonable and are quantitatively supported well via counterfactual evaluation.  Evaluation for attribution approaches is difficult since there exists no ground truth label. The counterfactual evaluation metrics proposed in this paper (inspired via TRAK [2] and DataModels [3]) will be useful for future research.
4. The writing quality of the paper is good. It was easy to follow the main contributions of the paper and understand the background of data attribution.

[1] https://www.klgates.com/Recent-Trends-in-Generative-Artificial-Intelligence-Litigation-in-the-United-States-9-5-2023
[2] Park, Sung Min, et al. "Trak: Attributing model behavior at scale." _arXiv preprint arXiv:2303.14186_ (2023).
[3] Ilyas, Andrew, et al. "Datamodels: Predicting predictions from training data." _arXiv preprint arXiv:2202.00622_ (2022).

**Weaknesses:**

1. A limitation of the proposed approach is the fact that attribution scores are only provided for a single denoising step. This is unintuitive, as it requires multiple steps to be analyzed to understand how an image was generated. It would be good to obtain a single-shot attribution score for the entire diffusion trajectory. While simple heuristics can be employed to obtain this from the current approach, it's unclear if these are useful and interpretable.

2. There is little analysis regarding how attributions change throughout the diffusion trajectory. It would be interesting to analyze more how ranking b/w attributions stay consistent b/w different timesteps. For example, what's the correlation coefficient b/w attributions of two timesteps close to each other v/s further away? How many of the +ve influence samples in the initial/middle timesteps stay positive throughout the diffusion trajectory?

3. The claim regarding conditioning likelihood increasing in small time intervals is a bit weak (i.e. features appear in specific timesteps). This should be more rigorously studied for multiple generated images on CIFAR-10 and MS-COCO.

Minor -
1. The font, and margins of this submission violate ICLR guidelines. This should be corrected in the next version of the paper.

**Questions:**

1. This is merely a suggestion and could help strengthen the paper. It would be interesting to compare attributions using a similar framework as [1] by fine-tuning large text-to-image models such as Stable Diffusion on a few images using a dreambooth-like approach. In this case, the attributions should have a higher +ve influence on the fine-tuning dataset. This can be done even for a small random subset of LAION images, instead of the entire dataset.  This can also be done for text-conditioned models on MS-COCO.

2. Several important details are missing regarding the attribution approach. TRAK uses a random projection matrix to compress gradients to low dimensional space, the dimension hasn't been mentioned at all. Is this random projection step not done for attributing diffusion models? Are multiple checkpoints used to estimate attribution scores? Are these trained on different subsets of training data? How much storage and compute is required for estimating the attribution scores?

[1] Wang, Sheng-Yu, et al. "Evaluating Data Attribution for Text-to-Image Models." arXiv preprint arXiv:2306.09345 (2023).

---

> ### Author Response · Authors · 2023-11-16
> **Rebuttal Response (1/2)**
>
> We thank the reviewer for the helpful feedback.
>
> ### Attributing individual steps rather than the full trajectory
> Indeed, within our framework we attribute at individual steps. In our view, attributing at either individual steps or the full trajectory have complementary benefits, and studying data attribution with respect to full diffusion trajectory is an important direction for future work. In particular, full-trajectory attribution might be more relevant towards matters of copyright violation and data valuation (as it might better assess how much each training image contributes to the final image). However, as we demonstrate in our work, step-specific attributions help us to gain a more fine-grained understanding of the way in which diffusion models use data, and can help to isolate attributions to individual features. For example, in Figure C.3 in Appendix C.2, we highlight cases in which the same image is highly influential at one step but negatively influential at another step. This shows that influences actually evolve over steps, which we could not have identified if we attributed the full diffusion trajectory as a whole.
>
> That said, we agree that attributing the full trajectory is an interesting and important future direction to extend our current work.
>
> ### How attributions change throughout the trajectory
> We recap some existing analyses in our paper to study how attributions change throughout the trajectory:
> 1. We identify a pattern for how attributions relate to the formation of features over steps. Specifically, we find that positive influencers begin to reflect a given feature as soon as the likelihood of a feature being in the final image begins to increase, while negative influencers begin to reflect a given feature when the likelihood of the feature reaches near 100% probability.
> 2. In Appendix C.2, as we mentioned above, we study the overlap between top and bottom influencers. In fact, we find that whether a given training image was one of the top most influencers does not correlate to whether it will be a negative influencer as well at another step.
>
> Following the reviewer’s suggestion, in Appendix C.2 we have added a plot of the correlation between attribution scores as a function of their proximity (Figure C.5). As expected, we find that the correlation between steps decreases as the distance between steps increases. Interestingly, we find that once attribution scores are computed at around 500 steps apart, their correlation reaches near zero. These findings again highlight the local nature of attributions.
>
> ### Likelihood increasing in small time intervals
> To address the reviewer’s concerns, we have added a new figure, Figure C.11, to Appendix C.5 that includes additional examples of the appearance of features over the steps of the diffusion trajectory. We would also like to point the reviewer to Figure C.2 in Appendix C.4. Here, we visualize the emergence of features in a sample generated by Stable Diffusion, and find that in this setting features appear often in a short interval of steps.
>
> Additionally, we would like to highlight that neither our method nor our evaluation strategy depend on this empirical phenomenon. We instead consider this finding as an avenue to bridge features of the synthesized images with intervals of the diffusion process, thus providing a more intuitive explanation of our attributions.
>
> ### Adhering to margin and font
> We thank the reviewer for pointing out the incorrect margins and font size in our submission. This was an unintentional accident that happened due to a git merge, and we fixed the revised version to adhere to the ICLR guidelines. The margins we used were wider than the ICLR guidelines, however it turns out our font was also wider. After fixing both the font and margins, we found that there was very little difference in the resulting paper length (about one paragraph). We apologize for this error.
>
> ### Fine-tuning text-to-image models
> We thank the reviewer for the interesting suggestion. We agree that it would be valuable to compare our method to the data attribution framework in [1], which identifies “ground truth” attributions by fine-tuning a text-to-image model on a given training image, guaranteeing that it is highly influential. Since the provided code for this work focused on attributions for Stable Diffusion, to compare to this method we would need to either adapt it to MS COCO models, as suggested, or implement our approach for subsets of LAION. Both of these tasks were not feasible during the rebuttal period, but we will consider implementing this comparison for the camera-ready version of the paper.

---

> > ### Author Response · Authors · 2023-11-16
> > **Rebuttal Response (2/2)**
> >
> > ### Missing Implementation Details
> > We appreciate the reviewer pointing out the missing details regarding the hyperparameters we use with TRAK. Below, we provide a summary, which we have also added to Appendix A of the paper:
> > - We do indeed use a random projection. In particular, we used a projection dimension of 4096 for CIFAR-10 and 16384 for MS COCO; it is worth noting that we achieved similar quantitative results by using a projection dimension of 2048.
> > - We do use multiple model checkpoints to estimate attribution scores. In particular, we use 100 model checkpoints for CIFAR-10, and 20 for MS COCO.
> > - We train the models that we use for estimating attributions on the entire training set
> > - The TRAK features for CIFAR-10 and MS COCO take up a total of ~20 GB of disk space. The compute time for obtaining a set of TRAK features is approximately proportional to training for one epoch, since we compute (and project) gradients for all samples in the training set.
> >
> > [1] Wang, Sheng-Yu, et al. "Evaluating Data Attribution for Text-to-Image Models." arXiv preprint arXiv:2306.09345 (2023).

---

> > > ### Author Response · Authors · 2023-11-22
> > > **Reminder**
> > >
> > > As the author-reviewer discussion period comes to a close, we'd like to ask if your comments have been sufficiently addressed by the rebuttal. We'd be happy to continue the discussion if you have further questions.
> > >
> > > Thank you again for your review!

---

### Official Review · Reviewer_Kccm · 2023-11-01

**Soundness:** 2 fair
**Presentation:** 1 poor
**Contribution:** 2 fair
**Rating:** 5
**Confidence:** 3

**Summary:**

This work attempts to explain the effect of training data points and their features on the synthesized images in a diffusion model. They suggest a framework to first define the notion of attribution for diffusion models and then study the validity of the attributions. The method is tested on two dataset along side two diffusion models (DDPM and LDM).

**Strengths:**

The choice of problem is a point of strength in this work. To understand the role of the training data in the final sampling result sheds light on the nature of the learned distribution and its properties. Also this question is tightly related to memorization and interpolation of training data which has practical implications such as data privacy.

**Weaknesses:**

The clarity of the writing can be significantly improved. The flow can be more streighforward. At its current form, the paper is too wordy to the extent that the main points are not clearly conveyed.

Relying on a classifier to find emergence of features that are important throughout the diffusion trajectory is problematic. The sharp increase in classification performance under Fig3 might be due to this particular classifier and dataset. For example, for a more nuanced dataset with images with more details, and more categories, the improvement in classification could become more gradual, due to the gradual appearance of large to fine features.

The results presented under figure 5 although show a significant effect on FID, it is not obvious what the effect shows and how it can be used. What is the relationship between the top k important training examples which are removed and the learned distribution or synthesis? How do we ensure that the increase in FID is not simply the effect of training on a smaller dataset?

In general, it is not clear to me what we learn from this result. The attribution qualitative results under fig 6 seem to me anecdotal. The results in Fig 7 are too low quality, so it is not easy to judge the effectiveness of the method.

**Questions:**

In order to reduce the cost of generating a distribution of generated conditional samples from $p(.|x_t)$, the conditional sampling is approximated with a one step denoising. We know that if the noise level is high (for larger values of t) the difference between iterative and single step denoising is significant. How is this approximation justified for the large sigma?

---

> ### Author Response · Authors · 2023-11-16
> **Rebuttal Response (1/2)**
>
> We thank the reviewer for the helpful feedback.
>
> ### Clarity of the submission
> To address the reviewer’s concerns over the clarity of the presentation, we have made a number of changes to the structure of the main text:
> - “The flow can be more straightforward”: We made revisions to Section 3 (which introduces our attribution framework) to simplify the language and make the presentation more modular. Additionally, we added an in-depth section on TRAK to the background section in order to better contextualize our method.
> - “the paper is too wordy”: we shortened Section 4 (which describes our method) and made more explicit connections to the preliminaries in Section 2.
>
>
> ### Use of Classifiers to Identify Features
> In our view, using a classifier is the only reasonable (and scalable) approach to identifying the emergence of features, as any other approach would likely involve manually inspecting images for their presence. However, the reviewer raises a valid concern about the robustness of our approach.
>
> To address the reviewer’s concerns, we have added an additional figure, Figure C.11, to Appendix C.5. In this figure, we visualize the emergence of a feature in the diffusion trajectory for three generated samples from the CIFAR-10 dataset using three different classifiers (ResNet trained on CIFAR-10, adversarially robust ResNet trained on CIFAR-10, and CLIP zero-shot initialized with the CIFAR-10 class names). We find that in each case, the outputs of all three classifiers are aligned.
>
> Additionally, we would like to point the reviewer to Figure C.2 in Appendix C.4. Here, we visualize the emergence of features in a sample generated by Stable Diffusion which has images with more details, and find that features still appear often in a short interval of steps.
>
> That said, we would like to highlight that neither our method nor our evaluation strategy depends on this empirical phenomenon. We instead consider this finding as a way to relate features of the synthesized images to specific steps of the diffusion process, thus providing a more intuitive interpretation of our per-step attributions.
>
>
> ### Interpretation of FID evaluation
> What the results of Figure 5 illustrate is that (as intended) the top-k most important training examples at step $t$ (as identified with our method) have a significant impact on the conditional distribution $p\_\theta(\cdot|x\_t)$ of generated images.
> Intuitively, we expect that the top-k (positively influential) images will share features with the original conditional distribution, but less so with the new conditional distribution (resulting after removing those top-k images).
>
> The increase in FID is not simply the effect of training on a smaller dataset, as we compare our method to the baseline of removing the top k training images according to either CLIP and pixel similarity. In this case, the model is indeed trained on a smaller dataset (with the same number of examples removed), yet the increase in FID is significantly less than the increase resulting from using our attribution scores.
>
>
> ### Evaluations of our Method
>
> We are not sure what aspects of Figure 7 the reviewer is referring to as low quality. In this figure, we show that even though our attribution scores are targeted at a specific timestep, we are still able to leverage them to attribute the full diffusion process, and do so much better than popular baselines of using image similarity for data attribution.
>
> Our patch-based attributions results (Figure 6) are only intended to convey qualitative results. In particular, this subsection of the experiments is intended to showcase a natural extension of our method, rather than a comprehensive evaluation of our framework.
>
> Importantly, we do not present these results (Figures 6, 7) as the primary evaluation in our work. Our main evaluation is presented In Section 5.2, where we report quantitative results for two metrics for evaluating attribution methods: linear datamodeling score (LDS) and distributional changes (as measured by FID) after removing top influencers. In both cases, our attributions significantly outperform the similarity baselines. Overall, our method is able to identify training examples most important to the generation of a given image, and moreover, precisely quantify the counterfactual impact of those training examples. We believe that these results sufficiently convey the efficacy of our approach.

---

> > ### Author Response · Authors · 2023-11-16
> > **Rebuttal Response (2/2)**
> >
> > ### The use of the approximation $\hat{x}\_0^{t}$
> > The reviewer raises a concern that the one-step estimate $\hat{x}\_0^{t}$ will significantly differ from samples from $p\_\theta(\cdot | x\_t)$ for large values of $t$. While this is indeed the case, we would like to highlight that, as we mention in the paper, $\hat{x}\_0^{t}$ is meant to approximate $\mathbb{E}\_{x\_0 \sim p\_{\theta}(\cdot|x\_t)}[x\_0]$, i.e., the *expectation* of $p(\cdot | x_t)$ rather than any individual sample from $p\_\theta(\cdot | x_t)$. Intuitively, the idea behind that is that we want to find which training examples steer the model towards generating the “expected” sample. We acknowledge that this is indeed a basic approximation to the full distribution. Yet, we nevertheless achieve counterfactually significant results when we evaluate the change (measured using the FID distance) in the full distribution $p(\cdot | x_t)$. Finally, we note that the approximation of  $\mathbb{E}\_{x_0 \sim p\_{\theta}(\cdot|x_t)}[x\_0]$ by $\hat{x}\_0^{t}$ holds for all steps $t$.

---

> > > ### Comment · Reviewer_Kccm · 2023-11-20
> > >
> > > Thank you for your response to my comments.
> > >
> > > I'm still not sure about the message of the paper. What is it that we are learning from applying your attribution framework on diffusion models? As I re-read the paper, I'm still trying to find the punchline. What is it that your method reveals about the data distribution or the synthesis trajectory? Can you state that message in one paragraph, e.g. in the abstract?
> > >
> > > There is the obvious observation that the features emerge in a coarse to fine manner in the trajectory. This effect has been known since the invention of diffusion models. One shot denoising solution is the posterior mean which results in the coarse to fine generation (due to constant spectrum of white noise, vs. $1/freq$ spectrum of natural images). Obviously, with larger noise, the posterior mean is taken over larger sets of clean images, which results in more ambiguous solutions (related to results in figures 1,2,3). So it is not clear what we learn about diffusion models by applying a classifier to such solutions. Similarly, it's not clear what the results in figures 5,6, and 7 elucidate, beyond the general and obvious observation that removing images of a particular type hurt the learning of the prior over images.
> > >
> > > Overall, I think this method can be useful to understand diffusion models and priors with more work, but the results need to be improved. Currently, it seems to me that they are too preliminary and it's not clear what conclusion we can draw from them.

---

> > > > ### Author Response · Authors · 2023-11-22
> > > > **Response**
> > > >
> > > > We thank the reviewer for engaging with our rebuttal. Below, we first address the reviewer’s question about our counterfactual results in Figures 5, 6, and 7. Then, we address the remaining questions in order.
> > > >
> > > > ### Meaning and significance of counterfactual evaluation
> > > > The reviewer claims that the results presented in Figures 5, 6, and 7 present a “general and obvious observation that removing images of a particular type hurt the learning of the prior over images”. Indeed, the goal of these experiments is precisely to show that the removal of certain (influential) training examples prevents the model from generating a given distribution of images. The main challenge (and the main goal of our paper) is to **efficiently find such examples**. As we present in these figures, our method uncovers training examples whose removal from the training data significantly changes the generations of the model (coincidentally, this is why we deem those training examples to be influential). Note that identifying such counterfactually significant images is not an “obvious” task: for example, removing the most similar images (according to either pixel-level similarity, or similarity in the latent space of a CLIP model) results in a much smaller change in the model generations.
> > > >
> > > > ### Broader goal of our framework
> > > >
> > > > As stated throughout the paper, the goal of our framework and method is to attribute the generated images of a diffusion model back to training data. More precisely, the aim is to identify how each training example increases or decreases the likelihood of a generated image. Leveraging the fact that the diffusion process is iterative, our approach is to attribute each step of the process, i.e., identifying the most influential training examples that influence a given point t.
> > > >
> > > > The reviewer seems concerned with the potential insights from applying this framework.
> > > > Note that the main contribution of our paper is formalizing the data attribution problem (in the context of diffusion models) and presenting a method; this is an important problem in of itself as we highlight several applications in the introduction. By applying our framework, we also uncover some new insights, for example that the influential training images change significantly over the diffusion trajectory. While we expect that our method can be used to extract further insights about the diffusion trajectory or the generated distribution, we believe that is beyond the scope of this work.
> > > >
> > > > We will clarify our message further in the abstract and other relevant places.
> > > >
> > > >
> > > > ### Features generated over time
> > > > The reviewer mentions that “obviously [...], features emerge in a coarse to fine manner [...] related to Figures 1, 2, and 3”. Indeed, this phenomenon can be inferred from Figures 1-3; however, we are not presenting this observation as our finding, only as illustration to accompany our main result: figure 1 presents our attribution scores across diffusion steps; figure 2 is part of the preliminaries and is only included for illustrative purposes; figure 3 shows that often features appear in a remarkably small interval of steps---note that this is a distinct observation from the folklore notion that “features emerge in a coarse to fine manner”.
> > > >
> > > >
> > > > ### Applying a classifier to samples from $p\_\theta(\cdot|x\_t)$
> > > > The reviewer states that “[...] it is not clear what we learn about diffusion models by applying a classifier to such solutions”. First, we would like to clarify that the image classifier used in the “Connecting features to specific steps.” paragraph of Section 3.1 is **not** part of our method. Second, we would like to reiterate that the goal of this experiment is to show that often the generation of a feature (e.g., the presence of a horse) happens in a remarkably short interval of steps during the diffusion process; this is an observation independent from our framework and method. Third, note that we are applying the classifier to samples from the final distribution (which are natural images), not the “blurry” images from the trajectory; we only use the latter to compute our attributions.
> > > >
> > > > Finally, the reviewer claims that our results are “too preliminary”. We would kindly ask the reviewer to provide 1) concrete areas where the reviewer believes our work falls short, and 2) specific suggestions for what the reviewer thinks needs improvement.
> > > >
> > > > We hope this addresses all the questions raised by the reviewer. As the author-reviewer discussion period comes to a close, we'd like to ask if your comments have been sufficiently addressed by the rebuttal. We'd be happy to continue the discussion if you have further questions.

---

### Author Response · Authors · 2023-11-16
**General Response**

### Writing
We thank the reviewers for their helpful feedback on the clarity of our writing. We address each reviewer’s concerns individually in each of the reviews. In summary, as a response to the reviewers’ suggestions, we have:
- Expanded the background on TRAK [1] in the preliminaries (Section 2)
- Simplified the language in Section 3 where we introduce our attribution framework
- Shortened Section 4 where we describe our method
- Moved figures closer to where we reference them in the text

### Experiment that suggests features appear in a short interval of diffusion steps
We thank the reviewers for engaging with our empirical observation that visual features often appear within a short interval of steps within the diffusion process (Figure 3). We would like to summarize a few key points regarding this:
- We would like to highlight that neither our method nor our evaluation strategy depends on this empirical phenomenon. We instead consider this finding as a way to relate features of the synthesized images to specific steps of the diffusion process, thus providing a more intuitive interpretation of our per-step attributions.
- Reviewers raised valid concerns about the robustness of this phenomenon with respect to both the number of diffusion trajectories being analyzed and the choice of the classifier. To address that, we provide additional evaluations (see Figure C.11) on multiple diffusion trajectories using a diverse set of image classifiers.

[1] Park, S. M., Georgiev, K., Ilyas, A., Leclerc, G., & Madry, A. (2023). TRAK: Attributing model behavior at scale. arXiv preprint arXiv:2303.14186.

---

### Meta-Review · Area_Chair_zAuu · 2023-12-08

**Metareview:**

A) This paper tackles the problem of attributing which training examples were used to create a particular image for diffusion models.

B) All reviewers agree that this is an important problem with the current legal + technical landscape. The reviewers agree that the proposed approached is novel and sound and lots of details are provided to reproduce the method.

C) Reviewers point out that the quality of writing can be improved and the way the results are presented can be confusing. Reviewers also raised a mismatch between the motivation and actual experiments which only give attribution score for a single step of the diffusion trajectory. Furthermore I find the lack of any comparisons to [1] and lack of any evaluations with real datasets that diffusion models trained on somewhat concerning. While it is nice the approach works on diffusion models trained on COCO and Cifar, the attribution problem only really appears for large web-scraped datasets. While the authors do not need to scale all the way to LAION-5B, there are many intermediate datasets that are in million example such as CC-12m, YFCC-15M, etc that can train high quality diffusion models [2]. I think the paper would be made much stronger with attribution experiments done on these web-scale datasets, I am not sure experiments on datasets with < 100k examples provide a lot of signal for the attribution problem. For this reason and others listed above I vote to reject this paper.


[1] Wang, Sheng-Yu, et al. "Evaluating Data Attribution for Text-to-Image Models.
[2] - Gu, Jiatao, et al. "Matryoshka diffusion models." arXiv preprint arXiv:2310.15111 (2023).

**Justification For Why Not Higher Score:**

The paper is promising but evaluation is lacking. Attribution is an important problem but must be tackled in a large scale.

**Justification For Why Not Lower Score:**

N/A

---

### Decision · Program_Chairs · 2024-01-16

Reject